# Inactivation of the Kv2.1 channel through electromechanical coupling

Ana I. Fernández-Mariño[1,4], Xiao-Feng Tan[1,4], Chanhyung Bae[1,4], Kate Huffer[1,2], Jiansen Jiang[3] & Kenton J. Swartz[1✉]

The Kv2.1 voltage-activated potassium (Kv) channel is a prominent delayed-rectifier Kv channel in the mammalian central nervous system, where its mechanisms of activation and inactivation are critical for regulating intrinsic neuronal excitability[1,2]. Here we present structures of the Kv2.1 channel in a lipid environment using cryo-electron microscopy to provide a framework for exploring its functional mechanisms and how mutations causing epileptic encephalopathies[3–7] alter channel activity. By studying a series of disease-causing mutations, we identified one that illuminates a hydrophobic coupling nexus near the internal end of the pore that is critical for inactivation. Both functional and structural studies reveal that inactivation in Kv2.1 results from dynamic alterations in electromechanical coupling to reposition pore-lining S6 helices and close the internal pore. Consideration of these findings along with available structures for other Kv channels, as well as voltage-activated sodium and calcium channels, suggests that related mechanisms of inactivation are conserved in voltage-activated cation channels and likely to be engaged by widely used therapeutics to achieve state-dependent regulation of channel activity.

Voltage-activated potassium (Kv) channels are critical for many physiological processes, including electrical signalling in neurons and muscle, neurotransmitter and hormone secretion, cell proliferation and migration, and ion homeostasis[8]. Kv channels are the largest family of ion channels in the human genome, with 40 members in 12 subfamilies identified following the cloning of the original Shaker (Kv1), Shab (Kv2), Shaw (Kv3) and Shal (Kv4) channels from *Drosophila melanogaster*[8]. Landmark structures of the Kv1 subfamily revealed a domain-swapped architecture between the peripheral S1–S4 voltage-sensing domains (VSD) and the S5–S6 segments that form the central pore domain (PD)[9–11], a feature also seen in structures of Kv3 (ref. 12) and Kv4 (ref. 13) channels, as well as in the related voltage-activated Na⁺ (Nav)[14] and Ca²⁺ (Cav)[15] channels. In these domain-swapped channels, the S4–S5 linker helices form a cuff around the PD, coupling movements of the positively charged S4 helix within each VSD to the opening of the internal S6 gate within the PD[9,10,16]. In response to sustained membrane depolarization, Kv channels inactivate, decreasing the flow of ions and influencing their contributions to electrical signalling and thus constituting a form of short-term memory[17,18]. Amongst Kv1 channels, inactivation can be fast (N-type in Shaker) and mediated by N-terminal domains that block the internal pore[19], or it can be slow (C-type in Shaker)[20] and caused by the dilation of the ion selectivity filter within the external pore to diminish ion permeation[11].

Hodgkin and Huxley were the first to measure delayed-rectifier K⁺ currents in the squid giant axon, and to demonstrate their critical role in action potential repolarization[21]. Although the Kv2.1 channel[22] is a prominent delayed-rectifier Kv channel in mammalian neurons[23], surprisingly it remains the only functional Kv channel for which no

structures have yet been reported. Two distinguishing features of the mammalian Kv2.1 channel are its delayed rectification and slow inactivation[22,24]. Slow inactivation in Kv2.1 can occur from both open and closed states, and recovery from inactivation is strongly voltage-dependent[24], yet its molecular mechanism remains elusive. Kv2.1 is widely expressed in pyramidal neurons in the cortex and hippocampus, where it has essential roles in regulating action potential shape, firing frequency and somatodendritic excitability[1,2]. Kv2.1 is also expressed in beta cells of the pancreas, where it regulates insulin secretion[25]. Mice in which Kv2.1 has been knocked-out are epileptic, hyperactive and have defects in spatial learning[26]. In humans, dominant mutations in Kv2.1 cause epileptic encephalopathy, characterized by developmental delays and epilepsy[3–7]. How these mutations alter the functional properties of Kv2.1 has been studied in only a few instances[3,4], and the lack of structures of Kv2.1 has limited our understanding of how they alter the functional mechanisms of the channel.

In the present study we report structures of the Kv2.1 channel in the membrane-like environment and investigate mutations causing epileptic encephalopathy in humans. One of these mutations within the internal S6 helix led to the identification of a previously unappreciated nexus of hydrophobic residues that is critical for inactivation and strategically positioned to be involved in coupling movements of the VSD to opening of the pore. Further functional and structural investigation of this 'hydrophobic coupling nexus' supports a mechanism of inactivation that results from dynamic alterations in electromechanical coupling between the voltage sensors and the pore, leading to the closure of the internal pore. The hydrophobic coupling nexus described here is conserved in Kv, Cav and Nav channels, suggesting

¹Molecular Physiology and Biophysics Section, Porter Neuroscience Research Center, National Institute of Neurological Disorders and Stroke, National Institutes of Health, Bethesda, MD, USA. ²Department of Biology, Johns Hopkins University, Baltimore, MD, USA. ³Laboratory of Membrane Proteins and Structural Biology, Biochemistry and Biophysics Center, National Heart, Lung, and Blood Institute, National Institutes of Health, Bethesda, MD, USA. ⁴These authors contributed equally: Ana I. Fernández-Mariño, Xiao-Feng Tan, Chanhyung Bae. ✉e-mail: swartzk@ninds.nih.gov

that this mechanism of inactivation is likely to be common to other voltage-activated cation channels.

## Kv2.1 structure in lipid nanodiscs

We expressed mVenus-tagged rat Kv2.1 channels containing residues 1–598−a minimum Kv2.1 construct displaying functional properties similar to the full-length protein (Fig. 1a and b)−in mammalian cells, reconstituted the protein into lipid nanodiscs (Extended Data Fig. 1a–d) and solved the structure using cryogenic-electron microscopy (cryo-EM) (Extended Data Fig. 1e–h and Supplementary Fig. 1). We solved the structure of the transmembrane (TM) regions of Kv2.1 without modelling the intracellular domains containing the N and C termini because the resolution in these regions was limited. The overall resolution of the Kv2.1 TM regions was 2.95 Å, and the maps were relatively uniform and high quality throughout, with discernable density for most side chains within the TM region, enabling model building excluding residues in the S1–S2 and S3–S4 loops (Extended Data Fig. 1h and Extended Data Table 1).

The structure of Kv2.1 resembles those solved previously for other Kv channels[10–13] even though the sequence of these channels varies considerably. As seen before, each subunit in the tetrameric Kv2.1 channel is composed of TM helices (S1–S6) (Fig. 1c,d), with the S1–S4 helices from each subunit forming individual VSDs and the tetrameric arrangement of S5–S6 helices forming the central PD. As in the Kv1 channel structures[10,11], Kv2.1 displays a domain-swapped architecture with the S1–S4 VSDs positioned near the S5–S6 pore-forming helices of the adjacent subunit (Fig. 1c,d). The structure of the Kv2.1 solved in lipid nanodiscs exhibits extra densities corresponding to multiple phospholipid molecules that appear to be bound to the protein (Fig. 1c,d) similar to those seen in Kv1 and Shaker Kv channels[10,11].

## Voltage-sensing domains

Basic residues in the S4 helices of Kv channels sense voltage by moving outward in response to membrane depolarization as the inner side of the membrane becomes less negative and inward upon repolarization as the inside becomes more negative[27]. The outward movement of S4 can be measured as an ON gating current, whereas the inward movement of S4 can be measured as an OFF gating current[27]. Three residues in Kv2.1 (F236 and E239 in S2 and D262 in S3) are positioned similarly to the charge transfer centre described in Kv1 channels[28], a region within the VSD that stabilizes the S4 basic residues within the membrane (Extended Data Fig. 2a). The S4 helix of Kv2.1 contains five basic residues within the membrane-spanning portion of the helix (Extended Data Fig. 2a). Three of these (R296, R299 and R302) are positioned external to the charge transfer centre with K305 and the innermost R308 near the charge transfer centre (Extended Data Fig. 2a). The accessibility of most of these basic residues to the external solution, together with the observation that most of the gating charges have moved on depolarization to 0 mV[29] (Fig. 2d,e), suggests that the VSDs of Kv2.1 have been captured in an activated state.

Although no structures have been solved for domain-swapped Kv channels with the VSDs in a resting state, a key constraint on the extent of the S4 movement in Shaker Kv channels comes from disulfide bonds or Cd[2+] bridges that trap the VSDs in the resting state when Cys residues are introduced at the external end of S4 and near the middle of S1 or S2 (ref. 30). To explore whether the S4 helix of Kv2.1 moves to a similar extent as in Shaker, we introduced the equivalent Cys substitutions into S4 and S1 of Kv2.1 (Extended Data Fig. 3a) and explored whether disulfide or Cd[2+] bridges could form. Cells expressing the double Cys mutant exhibited voltage-activated currents that increased following treatment with the reducing agent dithiothreitol (DTT) and could be inhibited by treatment with the oxidizing agent Cu-Phenanthroline (Cu-Phe) (Extended Data Fig. 3b,c), suggesting that disulfide bonds

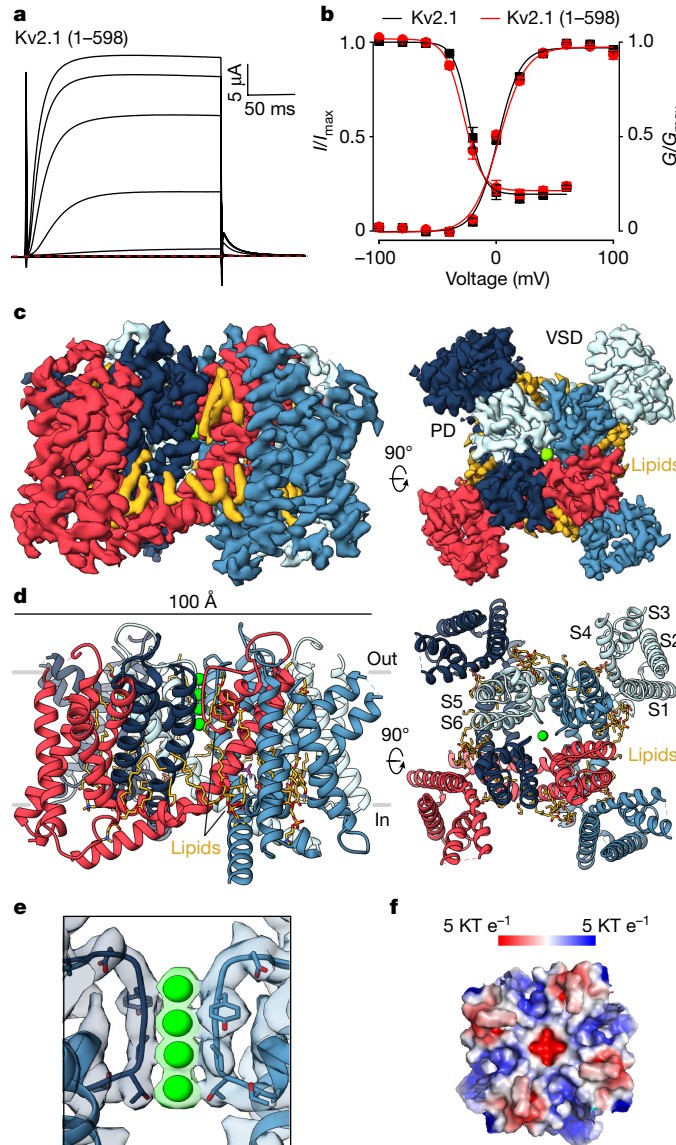

**Fig. 1 | Structure of the Kv2.1 channel. a**, Current traces for the structural construct of Kv2.1 (1–598) recorded in 2 mM external K[+] from −100 mV to +100 mV (40 mV increments) using a holding voltage of −90 mV and a tail voltage of −50 mV. Red dotted line denotes zero current. **b**, Normalized conductance–voltage (G–V) relations obtained using tail currents from traces like those in **a** and voltage–steady-state inactivation relations curves (I–V) obtained from a three-pulse protocol (Fig. 3d) comparing the structural construct (G–V, $V_{1/2} = 1.3 \pm 1.2$ mV, $z = 2.7 \pm 0.2$, $n = 6$ cells in two independent experiments; I–V (inactivation), $V_{1/2} = -27.8 \pm 0.9$, $z = 3.2 \pm 0.1$, $n = 6$ cells in two independent experiments) with the full-length Kv2.1 channel (G–V, $V_{1/2} = 0.8 \pm 1.8$ mV, $z = 2.8 \pm 0.4$, $n = 10$ cells in eight independent experiments; I–V (inactivation), $V_{1/2} = -23.3 \pm 1.0$, $z = 3.9 \pm 0.3$, $n = 3$ cells in independent experiments). Solid symbols represent mean and solid lines corresponds to fits of the Boltzmann Equations. Error bars denote the standard error of the mean (s.e.m.). **c,d**, Side and external views of the Kv2.1 EM map (**c**) and model (**d**), with each subunit shown in different colours. EM densities that could correspond to lipids are in yellow. **e**, Close-up view of the selectivity filter model superimposed with the EM map with the K[+] ion densities highlighted in green. **f**, Top view of the electrostatic surface of the extracellular PD of Kv2.1.

formed spontaneously, and could be broken with DTT. Following reduction with DTT, the double Cys mutant could be inhibited by Cd[2+], suggesting that metal bridges could form between the introduced Cys residues (Extended Data Fig. 3d,e). We also found that the introduction

of a single Cys in S4 was able to form disulfide or Cd²⁺ bridges with a native Cys in S2 (C232) positioned nearby to where we introduced a Cys in S1 (Extended Data Fig. 3f–m). Inspired by these results, we attempted to solve structures of the Cys mutants following treatment with Cu-Phe, Cd²⁺ or Hg²⁺. Although both constructs were successfully purified to homogeneity and reconstituted in nanodisc, imaged particles after treatment with Cu-Phe or metals did not show promising features in two dimensional (2D) class averages. Nevertheless, these findings suggest that the S4 helices in Kv2.1 move similar distances when compared to the Shaker Kv channel.

## Pore domain

Within the external pore where the ion selectivity filter resides, backbone carbonyl oxygens line the ion permeation pathway and are positioned similarly to what has been seen in other K⁺ channels[10,11,31] (Fig. 1e). There are also strong EM densities at four positions within the filter along the central axis (Fig. 1e), suggesting that the filter is occupied by ions at the four sites originally identified in a prokaryotic K⁺ channel[31], and probably represents a conducting conformation. Inspection of HOLE diagrams for the permeation pathway of Kv2.1 indicates that the internal pore of the channel is dilated to a similar extent as Kv1 channels captured in an open state[10] (Extended Data Fig. 2b). The external vestibule of the pore in Kv2.1 is unique among K⁺ channels in that it contains many basic residues (Fig. 1f and Extended Data Fig. 2c), explaining the insensitivity for Kv2.1 to cationic pore-blocking scorpion toxins[32].

## Epileptic encephalopathy mutations in Kv2.1

In humans, missense mutations in Kv2.1 cause epileptic encephalopathy, a disorder characterized by developmental delays in the first year of life preceding the onset of epileptic seizures[3–7]. Previous studies with four epileptic encephalopathy mutations within the PD of Kv2.1 revealed that the channel remains voltage-activated in one instance but loses K⁺ selectivity[3], whereas three other mutations appear to be non-functional[4]. To advance further study of the mechanisms of pathogenesis, we investigated the functional properties of 15 mutations causing epileptic encephalopathy at positions resolved in our structure of Kv2.1 (Fig. 2a,b and Extended Data Fig. 2d). We could not see functional activity for seven mutants located within the PD near the ion selectivity filter (S343R, T370I, G375R, G377R and P381T) or in the pore-lining S6 helix (C393F and G397R) (Extended Data Fig. 2d). That these mutations cause a marked loss-of-function can be rationalized because each introduces radical alterations in regions critical for ion permeation and produce the most severe disease[7]. Seven other mutations were functional with voltage-activation relationships shifted to negative voltages, indicating a gain-of-function, or to positive voltages, indicating a loss-of-function (Fig. 2b, Extended Data Fig. 2e and Supplementary Table 1). Among all the mutations tested, the F412L mutation (Fig. 2c) was particularly notable because it failed to conduct K⁺ over the wide range of voltages investigated regardless of the concentration of external K⁺ used. However, in this mutant we could measure robust gating currents resulting from the movement of charged S4 helix as it moves across the membrane electric field, including ON gating currents upon depolarization and OFF gating currents upon repolarization (Fig. 2d,e and Extended Data Fig. 2f,g). Although each of these epileptic encephalopathy mutations are intriguing, we were inspired to study the F412L mutant in greater detail because its non-conducting phenotype is reminiscent of a mutation in the Shaker Kv channel[33] that enabled the mechanism of slow C-type inactivation to be elucidated[11].

## A hydrophobic coupling nexus involved in inactivation

In the structure of Kv2.1, F412 is located at the internal end of the S6 helix and projects away from the ion permeation pathway towards the

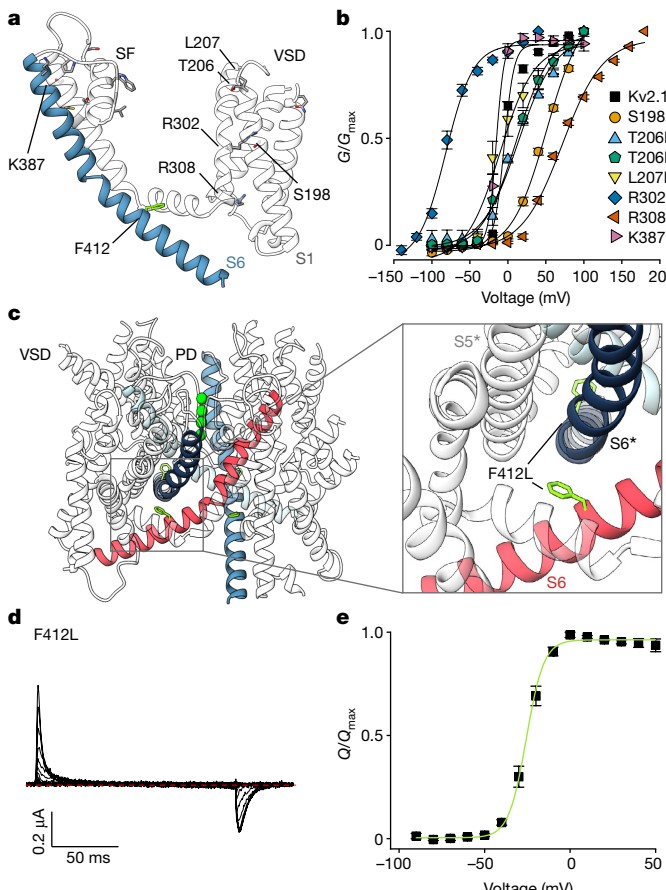

**Fig. 2 | Epileptic encephalopathy mutations in Kv2.1. a**, Mutations causing epileptic encephalopathy in humans mapped onto one monomer of Kv2.1, shown as a side view. **b**, $G$–$V$ relations recorded for epileptic encephalopathy mutations highlighted in **a** and obtained from a family of voltage steps ranging from −150 to +200 mV in 100 mM external K⁺. Symbols represent mean and solid curves correspond to a fit of the Boltzmann equation. See Extended Data Fig. 2e for traces and Supplementary Table 1 for parameters of the fits and $n$ values. **c**, Location of F412 in the Kv2.1 structure with adjacent S6 helices highlighted with different colours. **d**, Gating currents recorded for the F412L epileptic encephalopathy mutation from −90 mV to +50 mV in 2 mM external K⁺ (10 mV increments) from a holding voltage of −90 mV using a P/−4 protocol to subtract leak and capacitive currents. Red dotted line denotes zero current. **e**, Normalized $Q$–$V$ relation obtained for F412L by integrating the OFF gating currents. Symbols represent mean and green solid curve corresponds to a fit of the Boltzmann Equation with $V_{1/2}$ = −25.2 ± 0.4 mV, $z$ = 4.4 ± 0.2 ($n$ = 4 cells in two independent experiments). For all panels error bars are s.e.m.

peripheral S1–S4 VSD, where it forms a nexus with interacting hydrophobic residues that includes L316 in the S4–S5 linker of the same subunit, as well as L329 and L403 in the S5 and S6 helices, respectively, of the neighbouring subunit (Fig. 3a). Previous structural and functional studies of domain-swapped Kv1 channels[9,10,16,34] support the concept that the S4–S5 linker helix—which connects the S1–S4 VSDs to the PD—and how this helix packs together with the intracellular extension of the S6 helix, are critical for electromechanical coupling, wherein movements of the S4 can drive opening and closing of the internal pore. In Kv2.1, the hydrophobic interactions within the nexus involving F412 appear to be strategically positioned to couple movement of an individual S4 helix and S4–S5 linker to those of two pore-lining S6 helices (Fig. 3a) and therefore to contribute to the concerted final opening transition[35–37].

To explore the mechanism underlying the non-conducting phenotype for F412L and the role of this hydrophobic coupling nexus, we

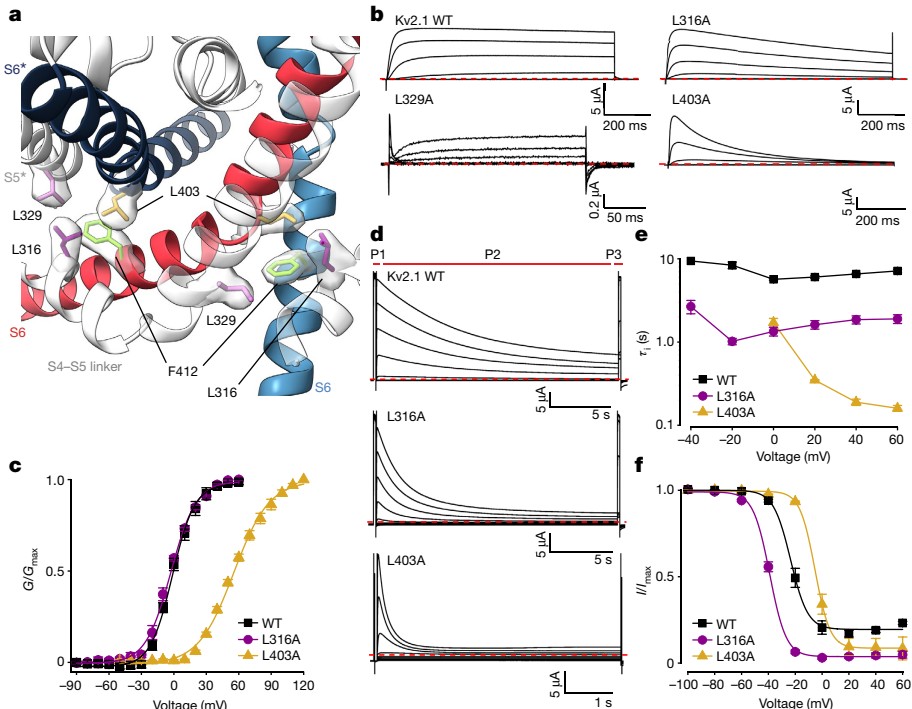

**Fig. 3 | A nexus of hydrophobic residues around F412 that are critical for inactivation. a**, A nexus of hydrophobic residues around F412 in the Kv2.1 structure with cryo-EM density for hydrophobic side chains. **b**, Current traces obtained for Kv2.1, L316A, L403A and L329A using 2 mM external K⁺ and P/−4 subtraction. Holding voltage was −90 mV, steps were from −90 mV to +60 mV (+50 mV for L329A) in 20 mV increments and tail voltage was −50 mV (−90 mV for L329A). **c**, G−V relations obtained from tail currents for Kv2.1 ($n$ = 13 cells in 10 independent experiments), L316A ($n$ = 3 cells in two independent experiments) and L403A ($n$ = 5 cells in two independent experiments) using 2 mM external K⁺. Holding voltage was −90 mV, voltage steps were 200 ms and tail voltage was −50 and −60 mV for Kv2.1 and L316A, respectively. For L403A, voltage steps were 20 ms and tail voltage was 0 mV. Symbols represent mean and smooth curves are fits of a Boltzmann equation (Kv2.1, $V_{1/2}$ = −1.7 ± 0.8 mV, $z$ = 2.6 ± 0.1; L316A, $V_{1/2}$ = −2.9 ± 0.8, $z$ = 2.2 ± 0.1; L403A, $V_{1/2}$ = 55.9 ± 1.3 mV, $z$ = 1.5 ± 0.1). **d**, Effect of Kv2.1 mutations on the rate and extent of inactivation assessed using a three-pulse (P1–P3) protocol with 2 mM external K⁺ and a holding

voltage of −100 mV. P1 was to +60 mV (+50 mV for L403A), followed by a brief step to −100 mV, P2 was from −100 to +60 mV for 20 s to allow channels to inactivate, and P3 was to the same voltage as P1 to assess the fraction of inactivated channels. P2 was 5 sec for L403A. **e**, Plot of time constants of inactivation ($\tau_i$) against P2 voltage. $\tau$ was obtained by fitting a single exponential function to the time course of the test current in P2. Data points are mean for Kv2.1 (black squares; $n$ = 3 cells in two independent experiments), L316A (purple circles; $n$ = 3 cells in two independent experiments) and L403A (yellow triangles; $n$ = 3 cells in three independent experiments). **f**, Fraction of non-inactivated channels during each P2 voltage step for Kv2.1 (black squares), L316A (purple circles) and L403A (yellow triangles) obtained by measuring the steady-state current at P3 normalized to P1. Same cells as in **e**. Smooth curves are fits of a Boltzmann equation (Kv2.1, $V_{1/2}$ = −23.3 ± 1.0, $z$ = 3.9 ± 0.3; L316A, $V_{1/2}$ = −39.0 ± 0.5, $z$ = 4.1 ± 0.2; L403A, $V_{1/2}$ = −5.2 ± 0.5, $z$ = 4.5 ± 0.2). For all panels error bars are s.e.m.

mutated the three interacting Leu residues and characterized their functional properties. The phenotype observed for the L329A mutation was similar to F412L in that we observed only small ionic currents along with measurable gating currents (Fig. 3b). Unexpectedly, both L316A in the S4–S5 linker and L403A in S6 remained conducting, yet with markedly accelerated inactivation (Fig. 3b,d,e). Both mutants speed the onset of inactivation, increase the extent of inactivation at steady-state and alter the voltage range over which steady-state inactivation occurs (Fig. 3d,e,f). The L316A mutation did not appreciably alter the voltage-activation relationship, suggesting a specific effect on inactivation, whereas L403A also produced a shift to more positive voltages (Fig. 3c). Consideration of these findings led us to hypothesize that the hydrophobic coupling nexus involving F412 is involved in coupling movements of S4 to opening and closing of the S6 gate, and that inactivation in Kv2.1 may result from alterations in coupling of the voltage sensor and gate, leading to closure of the internal pore even though the voltage sensors remain activated.

The non-conducting phenotype we observed in the F412L mutant in Kv2.1 is reminiscent of that observed for the W434F mutant of Shaker, which is non-conducting because the mutant strongly promotes C-type inactivation[11,33,38]. In structures of Shaker, W434 hydrogen bonds with D447 to stabilize a conducting state, and this interaction is broken

in the W434F mutant[11,38]. Residues equivalent to W434 and D447 are conserved in Kv2.1 (W365 and D378) and our structure of the conducting state of Kv2.1 suggests that they probably form a hydrogen bond (Extended Data Fig. 4a). To test whether inactivation in Kv2.1 is related to C-type inactivation in Shaker, we made the W365F mutation in Kv2.1 and observed that the mutant remains conducting (Fig. 4a) and notably inactivates more slowly and less completely than the WT channel, regardless of the concentration of external K⁺ (Extended Data Fig. 4b,c), suggesting that inactivation in Kv2.1 is probably distinct from C-type inactivation in Shaker. Other mutations that strongly influence C-type inactivation in Shaker have also been reported to not influence inactivation in Kv2.1 (ref. 39).

We next asked whether the intracellular S6 gate can open in F412L. In the non-conducting Shaker W434F mutant, movement of the intracellular gate can be detected using internal tetraethylammonium (TEA), which only enters the pore once the gate has opened upon depolarization and then slows closure of the gate and OFF gating currents upon repolarization (Fig. 4b,c)[40]. Although internal TEA slows deactivation of WT Kv2.1 (Extended Data Fig. 4d,e), it has no discernible effect on the similar OFF gating currents measured in F412L (Fig. 4b,c), suggesting that the internal pore of F412L remains closed to the blocker. As a control we tested whether internal TEA could slow gating currents

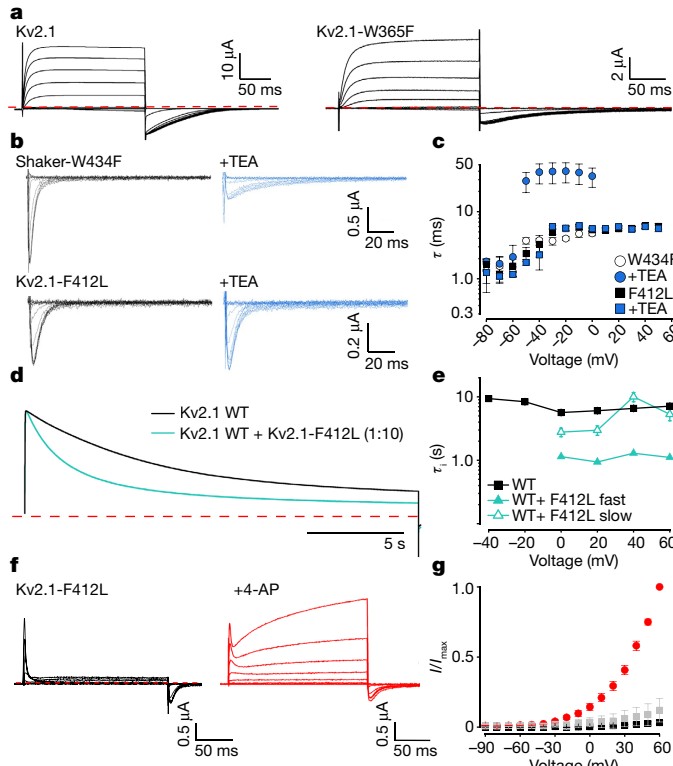

**Fig. 4 | Status of the internal gate in the F412L mutant of Kv2.1. a,** Current traces for Kv2.1 (left) and W365F (right). The holding voltage was −90 mV, depolarizations were from −100 to +100 mV (20 mV increments) and tail voltage was −50 mV. External K⁺ was 100 mM. Red dotted line denotes zero current. **b,** OFF gating currents recorded for Shaker W434F and Kv2.1 F412L in control (black) or after application of internal TEA (blue) using P/−4 subtraction and 2 mM external K⁺. The holding voltage for Shaker W434F was −100 mV, test depolarizations were from −100 to 0 mV (10 mV increments). For Kv2.1 F412L the holding voltage was −90 mV, test depolarizations were from −90 to +50 mV (10 mV increments). **c,** Time constants ($\tau$) for single exponential fits of the decay of OFF gating current in the absence or presence of internal TEA for Shaker W434F ($n = 5$ cells in five independent experiments) and Kv2.1 F412L ($n = 5$ cells in five independent experiments). **d,** Normalized ionic currents recorded for co-expression of Kv2.1 and the Kv2.1-F412L mutant with 2 mM external K⁺ at 60 mV. **e,** Plot of time constants ($\tau_i$) of inactivation against test voltage. $\tau$ values were obtained by fitting single (Kv2.1) or double (Kv2.1+Kv2.1-F412L) exponential functions to the time course of the test current in **d**. Data points are mean for Kv2.1 (black squares; $n = 3$ cells in two independent experiments) and Kv2.1+Kv2.1-F412L (green triangles; $n = 4$ cells in two independent experiments). **f,** Current traces recorded for Kv2.1 F412L before and during application of 10 mM external 4-AP. External K⁺ was 2 mM. Holding voltage was −90 mV, test depolarizations (200 ms) were from −90 to +60 mV (10 mV increments). **g,** Normalized steady-state current–voltage ($I–V$) relations for Kv2.1 F412L in control (black squares, $n = 6$ cells in two independent experiments), in 4-AP (red circles, $n = 6$ cells in two independent experiments) or after removal of 4-AP (grey squares, $n = 3$ cells in two independent experiments). Currents were measured at the end of the test depolarization and normalized to the maximum value obtained in the presence of 4-AP. For all panels error bars are s.e.m.

of the L329A mutant that displays both gating and ionic currents, the latter of which suggests that the gate can open to some extent. In the case of L329A, internal TEA inhibited the ionic currents and slowed OFF gating currents elicited by stepping from positive to negative voltage near the equilibrium potential for K⁺ (Extended Data Fig. 4f,g), suggesting detectable opening of the S6 gate in that mutant. From these results we conclude that F412L is non-conducting because the internal S6 gate is closed.

We next sought to establish a link between the non-conducting phenotype of F412L and the enhanced inactivation seen in mutations of the Leu residues in the hydrophobic coupling nexus. First, we reasoned that if F412L is non-conducting because it greatly speeds entry into an inactivated state, heteromeric channels comprised of WT and F412L might remain conducting but with more rapid inactivation, as previously observed for the W434F mutation in the Shaker Kv channel[33]. Indeed, when F412L is co-expressed along with the WT Kv2.1 channel we can readily detect a fraction of channels that inactivate considerably faster than the WT channel (Fig. 4d,e and Extended Data Fig. 4h–m), consistent with the dominant phenotype of the F412L mutation in humans[7]. Second, we used a pharmacological approach with the Kv channel inhibitor 4-aminopyridine (4-AP), which interferes with both opening and inactivation[41]. Critical determinants for 4-AP sensitivity are located with internal regions of the S6 helix[42–45] near the hydrophobic coupling nexus. When applied to Kv2.1, 4-AP partially inhibits channel currents, slowing activation and shifting the voltage-activation relationship to more positive voltages (Extended Data Fig. 5a–c), consistent with stabilization of a closed state. Importantly, 4-AP also interferes with inactivation of Kv2.1, slowing the onset of inactivation and diminishing the extent of inactivation at steady-state (Extended Data Fig. 5d–f). In the case of L316A, 4-AP slowed the onset and decreased the extent of inactivation (Extended Data Fig. 5g–i). In the case of L403A, the extent of inactivation was markedly reduced by 4-AP even though a fraction of channels inactivated with similar kinetics to that observed in control solutions (Extended Data Fig. 5j–l), presumably because this mutant weakens 4-AP binding[45] and these channels are not fully occupied by 4-AP at the concentration tested. Collectively, these results suggest that 4-AP similarly interferes with inactivation for the WT and Leu mutant channels. As a final test for whether the non-conducting phenotype observed in F412L is related to inactivation, we investigated the effects of 4-AP on F412L and observed that the compound actually rescued ion conduction, and that this notable effect was reversible after removal of 4-AP (Fig. 4f,g). Taken together, these findings support the hypothesis that the hydrophobic nexus has a critical role in inactivation in Kv2.1 and that the mechanism of inactivation involves an alteration in the coupling between the voltage sensors and the pore, leading to closure of the internal pore.

## Structure of the rapidly inactivating L403A mutant

To understand the structural basis of Kv2.1 inactivation involving closure of the internal pore, we attempted to solve the structure of the F412L mutant; however, collected cryo-EM images of the mutant showed no 2D classes with features of the Kv2.1 channel. We then pivoted toward the L403A mutant that promotes inactivation. The L403A mutant behaved better biochemically (Extended Data Fig. 1i) and the functional properties of the mutant in the Kv2.1(1–598) structural construct were similar to those in the full-length channel (Fig. 5a,b). We were able to collect a large dataset where about 80% of particles were classified to the activated-open state, while about 20% adopted a unique conformation where the internal pore appeared closed in two of the three dimensional (3D) classes (Supplementary Fig. 2). Refinement of one of these 3D classes with a closed pore using C1 symmetry resulted in a cryo-EM map with an overall resolution of 3.3 Å, from which we built a model for TM regions of the tetrameric channel (Fig. 5c and Extended Data Fig. 1j–m).

The structure of the individual VSDs and the ion selectivity filter in the L403A structure are indistinguishable from the structure of Kv2.1 that has activated voltage sensors and a conducting ion selectivity filter (Fig. 5c,d and Extended Data Fig. 6a,b). However, elsewhere the four subunits do not adopt the same conformation, most notably in the internal regions of the pore, including the S4–S5 linkers and S6 helices (Fig. 5e–h and Extended Data Fig. 6a,b). For the subunit exhibiting a conformation differing the most from the open state, which we will

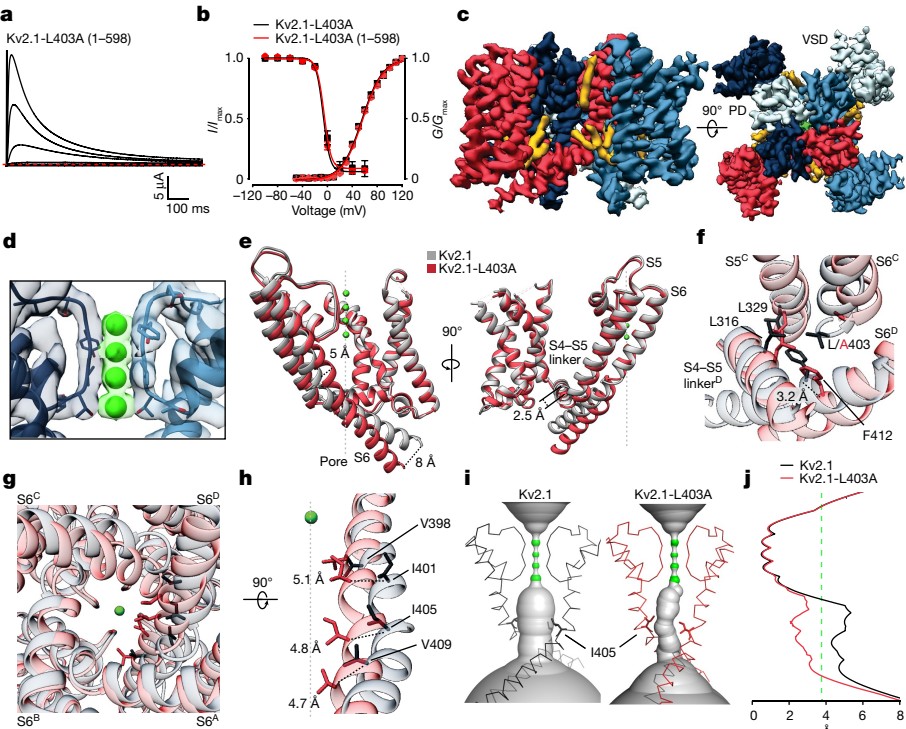

**Fig. 5 | Structural basis of inactivation in Kv2.1 channels. a**, Current traces for the structural L403A construct of Kv2.1 (1–598) recorded in 2 mM external K⁺ from −100 mV to +60 mV (20 mV increments) using a holding voltage of −100 mV.

Red dotted line denotes zero current. **b**, Conductance–voltage (*G–V*) relations and voltage–steady-state inactivation relations (*I–V*) obtained from a three-pulse protocol (Fig. 3d) comparing the L403A Kv2.1 (1–598) (*G–V*, $V_{1/2}$ = 56.7 ± 1.2 mV, $z$ = 1.5 ± 0.1, $n$ = 5 cells in two independent experiments; *I–V* (inactivation), $V_{1/2}$ = −6.9 ± 0.7, $z$ = 4.6 ± 0.1, $n$ = 7 cells in two independent experiments) with the L403A mutant in the full-length Kv2.1 channel (Fig. 3c,f). *G–V* relations were obtained from tail currents using a holding voltage of −90 mV, 20 ms voltage steps to between −50 mV and +120 mV (10 mV increments) and a tail voltage of 0 mV. Solid symbols represent mean and solid lines corresponds to fits of the Boltzmann equation. Error bars are s.e.m. **c**, Side (left) and external (right) views of the Kv2.1-L403A EM map, with each subunit shown in different colours.

EM densities that could correspond to lipids are in yellow. **d**, Model of the ion selectivity filter superimposed with the EM map, with the K⁺ ion densities highlighted in green. **e**, Superimposition of the most inactivated subunit of the L403A mutant (protomer D) with one subunit of Kv2.1 illustrating conformational changes in S6, S5 and the S4–S5 linker. **f**, Conformational changes in the hydrophobic coupling nexus between Kv2.1 and the L403A mutant protomer D. **g**, Superimposition of Kv2.1 and the L403 mutant structures viewed from the intracellular side of the membrane with key residues in S6 shown in stick representation. **h**, Superimposed views of the S6 helices of Kv2.1 and the most inactivated subunit (protomer D) of the L403A mutant. **i**, HOLE representations of the ion permeation pathway for Kv2.1 and the L403A mutant with the backbone for S6 and the selectivity filter of the models shown for reference. **j**, Plot of pore radius along the ion permeation pathway with dashed green line at the radius of a hydrated K⁺ ion.

refer to as the most inactivated protomer (chain D), there are three important features of the S6 helix that are distinct from the open state. First, the kink observed at the conserved PXP motif (PIP, 404–406) in the open state is straightened in L403A as the S6 helix in this region bulges similar to a π-helix (407–411) and translates towards the central axis of the pore by about 5 Å, with the internal end of the S6 helix translating by 8 Å (Fig. 5e,g,h). Second, the structural change within the PXP motif involves a rotation of the S6 helix by about 45° that repositions I401 and I405 from interacting with the neighbouring helices to lining the ion permeation pathway, occluding the pore (Fig. 5g,j and Extended Data Fig. 6c,d). Third, although the interactions between the S4–S5 linker and S6 are notably similar between the open state and the most inactivated protomer, as S6 changes conformation the S4–S5 linker translates towards the central axis of the pore by about 2.5 Å (Fig. 5e). The structural change seen in this subunit disrupts the interactions with F412 and each of the Leu residues in the hydrophobic coupling nexus as F412 moves by about 3 Å away from the positions the Leu residues occupy in the open state structure (Fig. 5f). The neighbouring protomer (chain C) adopts a unique conformation in the L403A mutant that appears to be intermediate between the open state and the most inactivated protomer D, while the other two protomers are most similar to the open state (Fig. 5f,g and Extended Data Fig. 6a). That only

two promoters in the L403A structure show an appreciable structural change while all four subunits contain the mutation suggests that the mutation alters the energetics of inactivation rather than producing a non-native structural change. Although it is possible that additional subunits may adopt conformations like the most inactivated protomer, we note that models containing all four subunits in that conformation contains multiple side chain clashes (Extended Data Fig. 6e–h), suggesting either that this is unlikely or that compensatory structural changes must occur. Taken together, these findings suggest that the hydrophobic coupling nexus has a central role in Kv2.1 inactivation and that this mechanism results from structural changes in the internal end of the S6 helix and the S4–S5 linker that close the internal end of the pore.

## Discussion

The structure of the Kv2.1 and the hydrophobic coupling nexus highlighted by the F412L epileptic encephalopathy mutation have fundamental implications for understanding the gating mechanisms of this family of Kv channels, as well as other voltage-activated ion channels. Our findings establish that interactions in the hydrophobic coupling nexus are dynamic and that alterations in those interactions leads to the closure of the internal pore during inactivation. This mechanism of

inactivation is distinct from those previously elucidated in the Shaker Kv channel, which include rapid N-type inactivation[19] and slow C-type inactivation[11,20]. The structures of Kv2.1 and the L403A mutant illuminate how the pore-lining S6 helix changes conformation from a conducting to a non-conducting inactivated state while the VSDs are activated. We envision that a related inactivated conformation of the pore also exists when the voltage sensors occupy resting or intermediate states because Kv2.1 can inactivate from closed states and recovery from inactivation is hastened by hyperpolarization[24]. We imagine that return of the VSDs to a resting state would enable the reengagement of the hydrophobic coupling nexus, allowing recovery from inactivation and fully engaging electromechanical coupling so that subsequent membrane depolarization can open the channel. Although we see no evidence that the ion selectivity filter is directly involved in inactivation of the internal pore in Kv2.1 (Figs. 1, 4 and 5 and Extended Data Figs. 4 and 6), there is evidence that these two regions of the channel are coupled. For example, the T373A mutation at the base of the filter alters slow inactivation in Kv2.1 (ref. 46) and when combined with the rapidly inactivating L403A mutant, we observe much slower inactivation (Extended Data Fig. 7a–c). It is therefore likely that the filter and internal inactivation gate are coupled, conceivably through a network of interacting residues connecting these two regions (Extended Data Fig. 7d). In the future it will be fascinating to explore how the voltage sensors influence inactivation and how the ion selectivity filter and the internal inactivation gate are coupled.

The mechanism of inactivation we describe for Kv2.1 is likely to operate in other types of voltage-activated cation channels. Alignment of our Kv2.1 structures with those available for Kv, Nav and Cav channels reveals that residues in the hydrophobic coupling nexus are relatively well-conserved in all three families (Supplementary Fig. 3), in particular when we consider how residues in the nexus defined here interact with a much more extensive network of hydrophobic residues within S6, the S4–S5 linker and S5 (Extended Data Fig. 8a,b). Residues in the hydrophobic coupling nexus of Kv2.1 are fully conserved in open state structures of Kv1, Kv3 and Kv4 channels (Extended Data Figs. 8c–f and 9a), and mutation of key residues involved in inactivation of Kv2.1 cause similar alterations in the Shaker Kv channel[40,47]. Closure of the internal pore during closed state inactivation of Kv4 channels has been proposed[48,49], and a recent study on Kv4.2 channels identified distinct populations of channels with VSDs in an activated conformation and the internal pore closed[13] (Extended Data Fig. 9). Although the structural changes we observe in the S4–S5 linker and S6 helix are distinct from those where the internal pore is closed in Kv4.2 (Extended Data Fig. 9), the interactions we see between residues in the hydrophobic coupling nexus of Kv2.1 are conserved in the open state of Kv4.2, they rearrange as the internal pore adopts non-conducting conformations (Extended Data Fig. 9) and mutations in the hydrophobic coupling nexus in Kv4 channels influence inactivation[49]. Thus, our findings in Kv2.1 support the proposal that inactivation in Kv4 channels involves closure of the internal pore and motivates further work to explore the role of this inactivation mechanism in other Kv channels.

Our findings also provide insight into the long-sought mechanism of fast inactivation in Nav channels, which was envisioned to occur through a classical ball-and-chain mechanism[50]. The critical isoleucine, phenylalanine and methionine (IFM) motif within the linker between domains III and IV proposed to function as a blocking particle in Nav channels[51] was recently resolved in structures of Nav1.4 and shown to be positioned peripheral to the internal pore[14], challenging the ball-and-chain mechanism. Structural alignment reveals that the IFM motif actually inserts directly into the equivalent of the hydrophobic coupling nexus located at the interface between domains III and IV (Extended Data Fig. 10a,c) and would necessarily disrupt the interactions we see in the open state of Kv2.1. The conformations of the S4–S5 linker and S6 helices in inactivated states of Nav channels are also similar to what we see in the L403A mutant of Kv2.1 (Extended

Data Fig. 10c,d). Domain IV of the Nav channels has a critical role in fast inactivation[52–54], which can be rationalized because the IFM motif inserts into the hydrophobic coupling nexus between domains III and IV. Involvement of the hydrophobic coupling nexus between domains IV and I in fast inactivation could also explain how the non-inactivating WCW mutant within domain I in Nav1.4 channels interferes with inactivation[55], as this mutant involves residues equivalent to the hydrophobic coupling nexus between domains I and IV (Extended Data Fig. 10b,d). A role of the hydrophobic coupling nexus between domains IV and I can also explain how mutations in the S4–S5 linker from domain IV interfere with inactivation[56] even though this helix is positioned distant from where the IFM motif engages between domains III and IV (Extended Data Fig. 10c,d). Like Kv2.1, Nav channels can inactivate through closed states, recovery from inactivation does not involve transitions through the open state[57]—unlike recovery from rapid N-type inactivation in Shaker and Kv1 channels[58]—and recovery from inactivation is hastened at negative membrane voltages[57]. It seems likely that the mechanisms of inactivation in Nav channels involve dynamic changes in electromechanical coupling and closure of the internal pore through a mechanism related to that described here for Kv2.1. A related hydrophobic coupling nexus is also seen in Cav channels, mutations nearby strongly promote inactivation[59] and recent structures of those channels have voltage sensors in activated states and internal pores that are closed[15] (Extended Data Fig. 10e,f), raising the possibility that they inactivate through a mechanism related to that for Kv2.1. Many therapeutic drugs engage inactivation mechanisms to modulate the activity of voltage-activated channels in a state-dependent manner, as first described for dihydropyridines and Cav channels[60]. The type of inactivation mechanism described here may help to elucidate the mechanisms of action for many drugs targeting members of the larger family of voltage-activated cation channels.

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

## Methods

### Kv2.1 channel expression using Baculovirus and mammalian expression system

To produce the Kv2.1 channel for cryo-EM, WT and L403A mutant channels were cloned into the pEG vector in which EGFP was substituted with mVenus[61] and expressed in tsA201 cells (Sigma-Aldrich) using the previously published Baculovirus-mammalian expression system (Invitrogen) with minor modifications[62]. tsA201 cells were not authenticated and mycoplasma contamination was tested routinely. In brief, the P1 virus was generated by transfecting Sf9 cells (Thermo-Fisher) (approximately 2.5 million cells in a T25 flask with a vent cap) with 50–100 ng of fresh Bacmid using Cellfectin (ThermoFisher). After 4 to 5 days of incubation in a humidified 28 °C incubator, the cell culture medium was collected by centrifugation (3,000$g$ × 10 min), supplemented with 2% FBS, and filtered through a 0.45 μm filter to harvest the P1 virus. To amplify the P1 virus, approximately 500 ml of Sf9 cell cultures at approximately 1.5 million cells ml$^{-1}$ density were infected with 1–200 μl of the virus and incubated in a 28 °C shaking incubator for 3 days. The cell culture medium was then collected by centrifugation (5,000$g$ × 20 min), supplemented with 2% FBS, and filtered through a 0.45 μm filter to harvest the P2 virus. The volume of the P1 virus used for the amplification was determined by carrying out a small-scale amplification screening in which approximately 10 ml of Sf9 cell cultures at the same density were infected with different volume of the P1 virus and harvested after 3 days to transduce tsA201 cells and compare the expression level of Kv2.1 channels using mVenus fluorescence intensity. The P2 virus was protected from light using aluminium foil and stored at 4 °C until use. To express the Kv2.1 channel, tsA201 cells at approximately 1.5 million cells ml$^{-1}$ in Free-style medium with 2% FBS were transduced with 10% (v/v) of the P2 virus and incubated with a 37 °C CO$_2$ incubator. To boost the protein expression, sodium butyrate (2 M stock in H$_2$O) was added to 10 mM at approximately 16 h of posttransduction. The culture was continued at 37 °C in a CO$_2$ incubator for another 24 h, and the cells were harvested by centrifugation (5,000$g$ × 20 min) and frozen at −80 °C until use.

### Kv2.1 channel purification

Before extraction of Kv2.1 channels from tsA201 cells, membrane fractionation was carried out using a hypotonic solution and ultra-centrifugation. Cells were first resuspended in a hypotonic solution (20 mM Tris pH 7.5 and 10 mM KCl) with protease inhibitors (pepstatin, aprotinin, leupeptin, benzamidine, trypsin inhibitor and PMFS) using a Dounce homogenizer, incubated at 4 °C for approximately 30 min, and centrifuged at 1,000$g$ for 10 min to remove cell debris. The supernatant was ultracentrifuged for 1 h (45,000 rpm, Beckman Ti45 rotor) and collected membranes were stored at −80 °C until use. To purify Kv2.1 channels, the fractionated membranes were resuspended in an extraction buffer (50 mM Tris pH 7.5, 150 mM KCl, 2 mM TCEP, 50 mM n-dodecyl-β-D-maltopyranoside (DDM; Anatrace), 5 mM cholesteryl hemisuccinate Tris Salt (CHS; Anatrace) with the protease inhibitor mixture used above) and extracted for 1 h at 4 °C. The solution was clarified by centrifugation (12,000$g$ × 10 min) and incubated with Co-TALON resins (TaKaRa) at 4 °C for 1 h, at which point the mixture was transferred to an empty disposable column (Econo-Pac Bio-rad). The resin was washed with 10 column volume of Buffer A (50 mM Tris pH 7.5, 150 mM KCl, 1 mM DDM, 0.1 mM CHS and 0.1 mg ml$^{-1}$ porcine brain total lipid extract; Avanti) with 10 mM imidazole, and bound proteins were eluted with Buffer A containing 250 mM imidazole. The eluate was concentrated using 100 kDa cut-off Amicon Ultra Centrifugal Filter (Millipore) to approximately 350–450 μl and loaded onto a Superose6 (10 × 300 mm) gel filtration column (GE Healthcare) and separated with Buffer A. All purification steps described above was carried out at 4 °C or on ice.

### Lipid nanodisc reconstitution of the Kv2.1 channel

Lipid nanodisc reconstitution was performed following the previously published methods with minor modifications[63]. On the day of nanodisc reconstitution, the Kv2.1 channel purified by Superose6 in detergent was concentrated to approximately 1–3 mg ml$^{-1}$ and incubated with histidine tagged Membrane Scaffold Protein (MSP)-1E3D1 and 3:1:1 mixture of 1-palmitoyl-2-oleoyl-sn-glycero-3-phosphocholine (POPC; Avanti), 1-palmitoyl-2-oleoyl-sn-glycero-3-phospho-(1′-rac-glycerol) (POPG; Avanti) and 1-palmitoyl-2-oleoyl-sn-glycero-3-phosphoethanolamine (POPE; Avanti) for 30 min at room temperature. The mixture was transferred to a tube with Bio-Beads SM-2 resin (approximately 30–50 fold of detergent; w/w; Bio-Rad) and incubated at room temperature for approximately 3 h in the presence of TEV protease (prepared in-house) and 2 mM TCEP to remove N-terminal fusion protein including poly-histidine and mVenus tag. The reconstituted protein was loaded onto Superose6 column (10 × 300 mm) and separated using 20 mM Tris and 150 mM KCl buffer at 4 °C. The success of nanodisc reconstitution was confirmed by collecting separated fractions and running SDS-PAGE to verify the presence of Kv2.1 and MSP-1E3D1 bands at a similar ratio. Typically, optimal reconstitution required the incubation of the 1:10:200 or 1:10:400 molar ratio of tetrameric Kv2.1, MSP-1E3D1 and the lipid mixture.

### Cryo-EM sample preparation and data acquisition

Concentrated samples (3 μl) of Kv2.1 (1.7 mg ml$^{-1}$) or the L403A (4.2 mg ml$^{-1}$) mutant in nanodiscs were applied to glow-discharged Quantifoil grids (R 1.2/1.3 Cu 300 mesh). The grids were blotted for 2.5 s, blot-force 4 and 100% humidity, at 16 °C using a FEI Vitrobot Mark IV (Thermo Fisher), followed by plunging into liquid ethane cooled by liquid nitrogen. Images were acquired using an FEI Titan Krios equipped with a Gatan LS image energy Filter (slit width 20 eV) operating at 300 kV. For Kv2.1 (1–598), micrographs were acquired at the nominal magnification of ×130,000 using a Gatan K2 summit direct electron detection camera, resulting in a calibrated pixel size of 1.06 Å per pixel. The typical defocus values ranged from −0.5 to −2.0 um. Exposures of 10 s were dose-fractionated into 50 frames, resulting in a total dose of 71 e$^-$ Å$^{-2}$. The data collection was automated using the Leginon software package (v.3.6)[64]. A total of 8,064 micrographs were collected. For the L403A mutant, a Gatan K3 Summit direct electron detector was used to record videos in super-resolution mode with a nominal magnification of ×105,000, resulting in a calibrated pixel size of 0.415 Å per pixel. The typical defocus values ranged from −0.5 to −1.5 μm. Exposures of 1.6 s were dose-fractionated into 32 frames, resulting in a total dose of 48 e$^-$ Å$^{-2}$. Images were recorded using the automated acquisition program SerialEM (v.3.8.1)[65]. A total of 31,119 micrographs were collected from two data collection sessions.

### Image processing

For Kv2.1 (1–598), all processing was completed in RELION (v.3.0)[66]. The beam-induced image motion between frames of each dose-fractionated micrograph was corrected using MotionCor2 (ref. 67) and contrast transfer function (CTF) estimation was performed using CTFFIND4 (ref. 68). Micrographs were selected and those with outliers in defocus value and astigmatism, as well as low resolution (greater than 5 Å) reported by CTFFIND4 were removed. The initial set of particles from 300 micrographs were picked using Gautomatch (v.0.56) (https://www2.mrc-lmb.cam.ac.uk/research/locally-developed-software/zhang-software/#gauto) and followed by reference-free 2D classification in RELION. The good classes were then used as template to pick particles from all selected micrographs using Gautomatch. A total of 2,374,290 particles were picked and extracted with 2× downscaling (pixel size of 2.12 Å). The starting model was generated using 3D initial model (C1 symmetry) with good particles from 2D classification. The best particles (361,379 particles) were selected iteratively

by selecting the 2D class averages and 3D reconstructions (using C4 symmetry) that had interpretable structural features. After removing duplicate particles, the selected particles were re-extracted using box size 240 pixels without binning (pixel size of 1.06 Å). To get good reconstruction within the transmembrane domain, the re-extracted particles were further 3D classified with a transmembrane domain mask in C4 symmetry (73,548 particles selected), followed by 3D auto-refine and CTF refinement. After that, Bayesian polishing was performed, and bad particles were removed from polishing particles using 2D classification. The selected polishing particles were subjected to 3D auto-refine in RELION. The final reconstruction was reported at 2.95 Å for Kv2.1.

For the L403A mutant, all processing was completed in RELION (v.4.0)[66]. The micrographs were divided into two data subsets based on the collection session. The beam-induced image motion between frames of each dose-fractionated micrograph was corrected and binned by two using MotionCor2 (ref. 67). Contrast transfer function (CTF) estimation was the performed using CTFFIND4 (ref. 68). Micrographs were selected and those with outliers in defocus value and astigmatism, as well as low resolution (greater than 5 Å) reported by CTFFIND4 were removed. The initial set of particles from 300 micrographs were picked using Gautomatch and followed by reference-free 2D classification in RELION. The good classes were then used as template to pick particles from all selected micrographs using Gautomatch. A total of 4,592,944 and 8,108,046 particles were picked and extracted with 2× downscaling (pixel size of 1.66 Å) separately. After several rounds of 2D classification, the good particles were subjected to 3D classification with both C1 and C4 symmetry using the initial model generated from Kv2.1 WT dataset. The best particles from 3D reconstructions that showed structure features were selected and merged. After removing duplicate particles, the selected particles were re-extracted using box size 300 pixels without binning (0.83 Å per pixel). A total number of 1,485,964 particles were aligned to the centre using a single-class 3D classification with C4 symmetry, followed by 3D auto-refine, CTF refinement and Bayesian polishing. Bad particles were removed from polishing particles using 2D classification followed by 3D auto-refinement. The reconstruction that was similar to Kv2.1 WT was reported at 2.89 Å.

To further classify the particles, the particles were expanded from C4 to C1 symmetry (command: relion_particle_symmetry_expand), yielding 5,943,856 particles (1,485,964 × 4). These particles were submitted to 3D classification for 10 classes without image alignment. A mask around the S5 linker and S5–S6 helices was created and applied for the focus classification. Among 10 classes, five classes showed an open internal pore similar to the Kv2.1(1–598) construct, and 2 classes exhibited a closed pore and inverted symmetry with respect to each other. The better class (505,078) with a closed pore was selected and subjected to a single 3D classification with a TM region mask (skip alignment), followed by 3D auto-refinement (local search). The final reconstruction was reported at 3.32 Å for the L403A mutant.

## Model building and structure refinement

For Kv2.1 (1–598), model building was first carried out by manually fitting the transmembrane domain of Kv1.2–2.1 paddle chimera channel (PDB 6EBM) into the EM density map using UCSF Chimera (v.1.15)[69]. The model was then manually built in Coot (v.0.9.8.1)[70] and refined using real space refinement in PHENIX (v.1.19.1)[71] with secondary structure and geometry restraints. The final model was evaluated by comprehensive validation in PHENIX. For the L403A mutant, the Kv2.1 WT model was first fit into the cryo-EM density map and we then manually built the model in Coot. During the building of the model, we found the density of the masked map at the end of S6 helix in protomer D to be poor, so we first built the backbone into the density of the unsharpened map to ensure that the trace of the backbone was correct before assigning the side chains within the density of the masked map. Structural figures were generated using PyMOL (v.2.4.1) (https://pymol.org/2/support.html) and UCSF Chimera (v.1.15).

## Structural alignments

Sequence-independent structure-based alignments (Supplementary Fig. 3 and Extended Data Figs. 8–10) were obtained using Fr-TM-Align[72] as previously described[73]. Pore radii were estimated using the MDAnalysis package (v.2.4.0)[74] implementation of the HOLE program (v.2.2.005)[75], which reports the radius of the largest sphere that can fit in the pore without intersecting with a neighbouring atom.

## Electrophysiological recordings

For electrophysiological recordings, the full-length rat Kv2.1 channel[22] cDNA was cloned into the pBlueScript vector. The structural construct containing a deletion after residue 598 (Kv2.1-1–598) was cloned into the pGEM-HE vector[76]. The W434F mutant[77] of the Shaker Kv channel was studied in parallel to the non-conducting F412L mutant of Kv2.1 because it results in non-conducting channels by promoting C-type inactivation[33] of the ion selectivity filter[11,78,79]. The Shaker W434F construct also contains a deletion of residues 6-46 to remove fast N-type inactivation[19,80] and was also cloned into pGEM-HE and pBSTA[34] vectors. Mutagenesis was performed by Quickchange Lightning Kit (Agilent) using the full-length channel unless otherwise indicated. The DNA sequence of all constructs and mutants was confirmed by automated DNA sequencing. cRNA was synthesized using the T7 polymerase (mMessage mMachine kit, Ambion) after linearizing with Nhe-I (NEB) for pGEM-HE or Not-I (NEB) for pBlueScript and pBSTA.

Oocytes (stage V–VI) from female *Xenopus laevis* frogs (approximately 1–2 years old from Xenopus I) were removed surgically and incubated for 1 h at 19 °C in a solution containing (in mM): NaCl (82.5), KCl (2.5), MgCl$_2$ (1), HEPES (5) and pH 7.6 with NaOH and collagenase Type II (2 mg ml$^{-1}$; Worthington Biochemical). The animal care and experimental procedures were performed in accordance with the Guide for the Care and Use of Laboratory Animals and were approved by the Animal Care and Use Committee of the National Institute of Neurological Disorders and Stroke (animal protocol number 1253). Defolliculated oocytes were injected with cRNA and incubated at 17 °C in a solution containing (in mM): NaCl (96), KCl (2), MgCl$_2$ (1), CaCl$_2$ (1.8), HEPES (5), pH 7.6 (with NaOH) and gentamicin (50 mg ml$^{-1}$; GIBCO-BRL) for 24–72 h before electrophysiological recording. Oocyte membrane voltage was controlled using either with an Axoclamp-2A two-electrode voltage clamp (Axon Instruments, Foster City, CA) or an OC-725C oocyte clamp (Warner Instruments, Hamden, CT) controlled using a pClamp (10.7). Data were filtered at 1–2 kHz (8-pole Bessel) and digitized at 5–10 kHz. Microelectrode resistances ranged from 0.2–0.6 MΩ when filled with 3 M KCl. Oocytes were studied in 150 µl recording chambers that were perfused continuously with an extracellular solution containing (in mM): NaCl (98), KCl (2), MgCl$_2$ (1), CaCl$_2$ (0.3), HEPES (5) and pH 7.6 with NaOH. When other external K$^+$ concentrations were used, NaCl was replaced with KCl. Most experiments were undertaken in lower external K$^+$ to approximate physiological conditions and elevated external K$^+$ was used in some experiments where inward tail currents were measured to compare the gating properties of different mutants. CdCl, copper-phenanthroline (Cu-Phe), dithiothreitol (DTT) and 4-aminopyridine (4-AP; Sigma-Aldrich) solutions were prepared fresh daily and added to the external recording solution to the desired final concentration. For internal tetraethylammonium (TEA; Fluka Analytics) experiments (Fig. 4b,c and Extended Data Fig. 4d–g), an injection pipette was used to inject oocytes with 100 nl of a 200 mM TEA solution in 100 mM KCl. If we assume an oocyte volume of 500 nl, the final intracellular concentration of TEA would be approximately 40 mM. All experiments were performed using a continuous flowing external solution and were carried out at room temperature (22 °C). Leak and background conductances were subtracted for tail current measurements by arithmetically deducting the end of the tail pulse of each analysed trace. In most instances, Kv channel currents shown

are non-subtracted, but where indicated, a P/−4 leak subtraction protocol[81] was employed.

The Boltzmann equation was fit to $G$−$V$ and voltage-steady-state inactivation ($I$−$V$) relations to obtain the $V_{1/2}$ and $z$ values according to:

$$\frac{I}{I_{max}} = \left(1 + e^{-zF(V - V_{\frac{1}{2}})/RT}\right)$$

where $I/I_{max}$ is the fractional activation of tail currents for $G$−$V$ relations or $I$ measured in P3 divided by $I$ measured in P1 for steady-state inactivation relations, $z$ is the equivalent charge, $V_{1/2}$ is the half-activation voltage, $F$ is Faraday's constant, $R$ is the gas constant and $T$ is temperature in Kelvin. Time constants of inactivation were obtained by fitting a single or double exponential function to the decay of currents using the following equation:

$$f(t) = \sum_{i=0}^{n} A_i e^{-t/\tau_i} + C$$

where $A$ is the amplitude and $\tau$ is the time constant. All analyses of electrophysiological data conducted using Origin 2020.

## Sample size

Statistical methods were not used to determine the sample size. Sample size for cryo-EM studies was determined by availability of microscope time and to ensure sufficient resolution for model building. Sample size for electrophysiological studies was determined empirically by comparing individual measurements with population data obtained under differing conditions until convincing differences or lack thereof were evident. For all electrophysiological experiments, $n$ values represent the number of oocytes studied from between two and ten different frogs (indicated as independent experiments).

## Data exclusions

For electrophysiological experiments, exploratory experiments were undertaken with varying ionic conditions and voltage clamp protocols to define ideal conditions for measurements reported in this study. Although these preliminary experiments are consistent with the results we report, they were not included in our analysis due to varying experimental conditions (for example, solution composition and voltage protocols). Once ideal conditions were identified, electrophysiological data were collected for control and mutant constructs until convincing trends in population datasets were obtained. Individual cells were also excluded if cells exhibited excessive initial leak currents at the holding voltage (greater than 0.5 μA), if currents arising from expressed channels were too small (greater than 0.5 μA), making it difficult to distinguish the activity of expressed channels from endogenous channels, or if currents arising from expressed channels were too large, resulting in substantial voltage errors or changes in the concentration of ions in either intracellular or extracellular solutions.

## Randomization and blinding

Randomization and blinding were not used in this study. The effects of different conditions or mutations on Kv2.1 channels heterologously expressed in individual cells was either unambiguously robust or clearly indistinguishable from control conditions.

## Reporting summary

Further information on research design is available in the Nature Portfolio Reporting Summary linked to this article.

## Data availability

All data needed to evaluate the conclusions in the paper are present in the paper and/or the Supplementary Material. Maps of Kv2.1 and the L403A mutant have been deposited in the Electron Microscopy Data Bank (EMDB) under accession codes EMD-40349 and EMD-40350, respectively. Models of Kv2.1 and the L403A mutant have been deposited in the Protein Data Bank with accession codes 8SD3 and 8SDA, respectively. Additional datasets used in this study include Protein Data Bank accession codes 7SIP, 6EBM, 7SSV, 7PHH, 7UKG, 7UKE, 7UKF, 7UKC, 6AGF and 7UHG.

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

**Acknowledgements** We thank T.-H. Chang for outstanding assistance and support throughout the duration of this project. We thank A. Jara-Oseguera, M. Holmgren, M. Mayer and members of the Swartz laboratory for helpful discussion, and H. Wang from the NIH Multi-Institute Cryo-EM Facility (MICEF) for assistance in acquiring cryo-EM data. This work utilized the NIDDK Cryo-EM Core Facility, the NIH MICEF and computational resources of the NIH HPC Biowulf cluster (http://hpc.nih.gov), the Intramural programme of the National Institute of Neurological Disorders and Stroke, NIH, NS002945 (K.J.S.) and the Intramural programme of the National Heart Blood and Lung Institute, NIH, HL006238 (J.J.).

**Author contributions** A.I.F.M., C.B., X.T. and K.J.S conceptualized the project. X.T., C.B., A.I.F.M., K.H., J.J. and K.J.S. were responsible for the methodology. A.I.F.M., X.T., C.B., and K.H. were responsible for the investigation. Visualization was done by A.I.F.M., X.T., C.B., K.H. and K.J.S. J.J. and K.J.S. were in charge of funding acquisition. Project administration was done by A.I.F.M., X.T., C.B. and K.J.S. K.J.S. supervised the project. A.I.F.M., X.T. and K.J.S. wrote the original draft of the paper. A.I.F.M., X.T., C.B., K.H., J.J. and K.J.S. reviewed and edited the paper.

**Competing interests** The authors declare no competing interests.

**Additional information**
**Correspondence and requests for materials** should be addressed to Kenton J. Swartz.

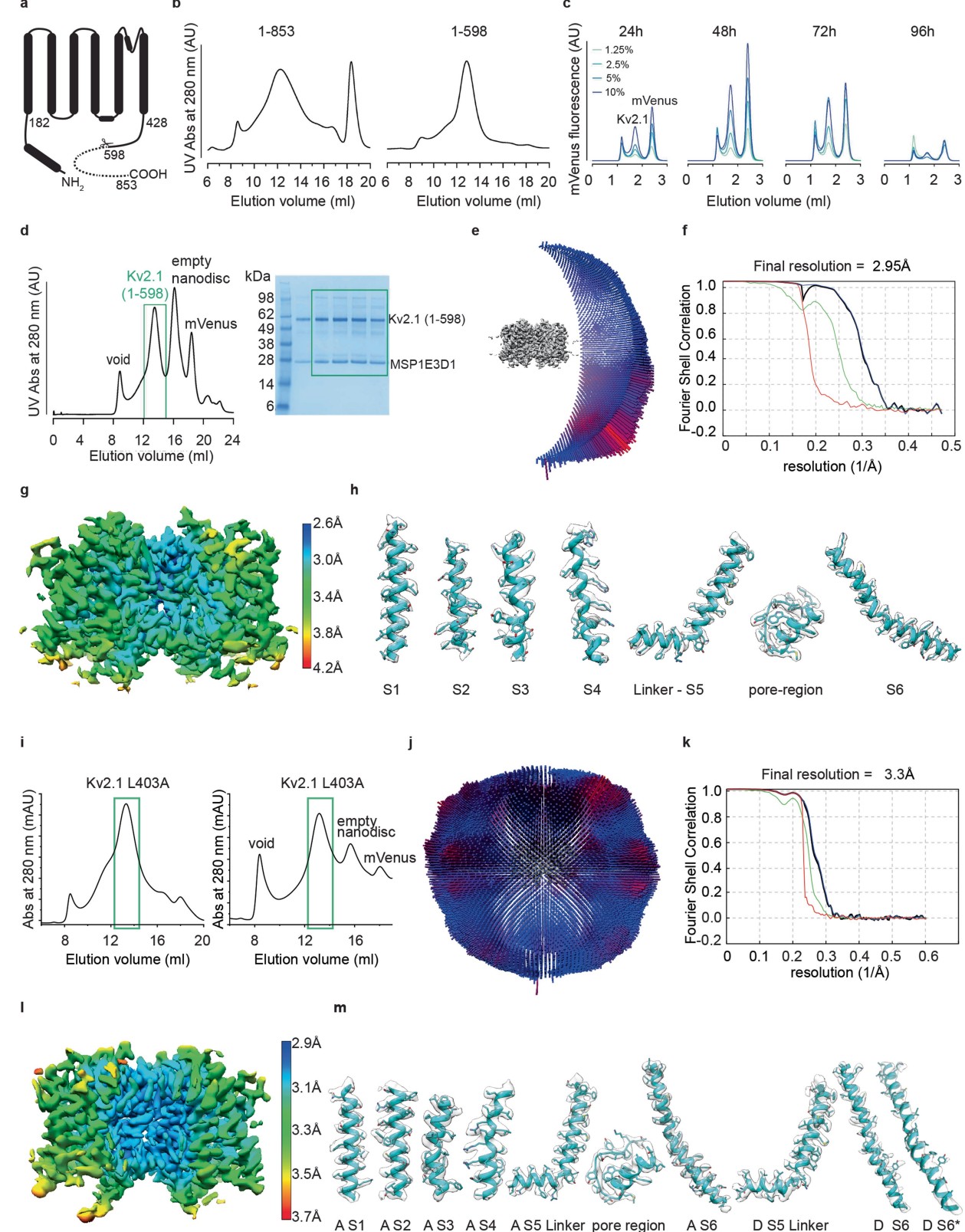

**Extended Data Fig. 1** | See next page for caption.

**Extended Data Fig. 1 | Biochemistry and cryo-EM imaging for the Kv2.1 channel. a)** Cartoon illustrating the Kv2.1 construct used for structure determination. **b)** Gel filtration chromatograms for the full-length Kv2.1 channel (1-853) and the 1-598 C-terminal truncated construct in detergent solutions using a superose6 increase (10/300) column. The full-length construct showed a broader peak, suggesting heterogeneity of the protein. **c)** Time-course for expression of the Kv2.1 (1-598) construct in tsA201 transduced with different baculoviruses titers (indicated as %) using mVenus fluorescence in gel filtration chromatograms (superose6 increase 5/150 column) to measure expression. **d)** Gel filtration profile (Superose6 increase; 10/300 column) for the Kv2.1 (1-598) construct after nanodisc reconstitution. Similar profiles were observed in over 7 independent experiments. The peak in the green box was collected and concentrated for preparing cryo-EM samples. SDS-PAGE analysis of the Kv2.1 (1-598) sample used for cryo-EM data collection. See Supplementary Information Fig. 4 for uncropped gel. **e)** Direction distribution plots of the 3D reconstruction illustrating the distribution of particles in different orientations with red indicating a larger number of particles. **f)** Fourier Shell Correlation (FSC) curves: green: rln FSC Unmasked Maps; blue: rln FSC Masked Maps; black: rln FSC Corrected; red: rln FSC Phase Randomized Masked Maps. **g)** Local resolution map for the entire TM region. **h)** Regional cryo-EM density for the TM regions of Kv2.1 (1-598). **i)** Gel filtration chromatograms for the L403A mutant of the 1-598 C-terminal truncated construct in detergent solutions (left). Superose6 increase (10/300) column was used for separation. The peak in the green box was collected and used for nanodisc reconstitution. Gel filtration profile (right; Superose6 increase; 10/300 column) for the L403A mutant of Kv2.1 (1-598) after nanodisc reconstitution. The peak in the green box was collected and concentrated for preparing cryo-EM samples. Similar profiles were observed in over 3 independent experiments. **j)** Direction distribution plots of the 3D reconstruction illustrating the distribution of particles in different orientations with red indicating a larger number of particles. **k)** Fourier Shell Correlation (FSC) curves: green: rln FSC Unmasked Maps; blue: rln FSC Masked Maps; black: rln FSC Corrected; red: rln FSC Phase Randomized Masked Maps. **l)** Local resolution map for the entire TM region. **m)** Regional cryo-EM density for the TM regions of the L403A mutant of Kv2.1 (1-598). The conformation of the A chain is similar to the activated/open conformation of Kv2.1 while that of the D chain is the most inactivated conformation of the mutant. D S6 shows masked cryo-EM density fitted with the model and D S6* shows unmasked cryo-EM density fitted with the model to further confirm that the trace of the backbone is correct.

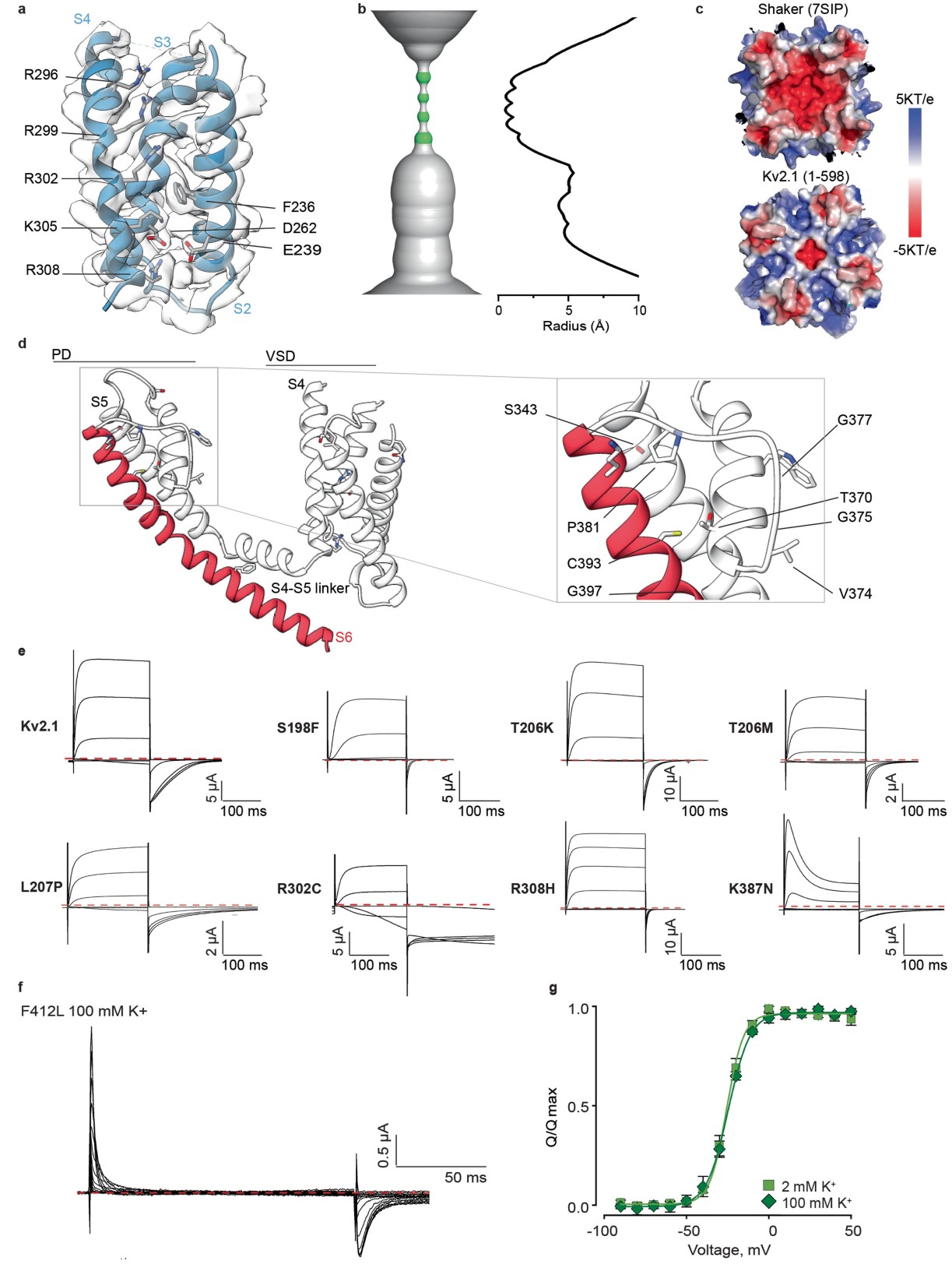

**Extended Data Fig. 2** | See next page for caption.

**Extended Data Fig. 2 | Conformation of the Kv2.1 structure and functional properties of epileptic encephalopathy mutations. a)** Side view of the model and EM maps for the S2, S3 and S4 helices within the VSD of Kv2.1. Basic residues in S4 and residues in the charge transfer center (F236, E239 and D262) are shown as sticks. **b)** Hole representation of the ion permeation pathway of the Kv2.1 structure and plot of pore radius along the length of the pore. **c)** View of the electrostatic surface of the extracellular PD of the Shaker channel (PDB 7SIP) and Kv2.1 (1-598). **d)** Side view of a single monomer of Kv2.1 with epileptic encephalopathy mutations shown in stick representation. Regions contributing to the PD and VSD are indicated and an expanded view of the PD is also shown. **e)** Current traces obtained with 100 mM extracellular K$^+$ from cells expressing mutant Kv2.1 channels. For WT Kv2.1, S198F, T206K, T206M, L207P and K387N mutants, steps were from −100 mV to +100 mV in 40 mV increments, holding voltage was −90 mV and tail voltage was −50 mV. R302C currents were recorded using a protocol with a holding voltage of −120 mV, using a pre-pulse to −140 mV followed by a family of voltage steps from −140 mV to +80 mV in 40 mV increments and a tail voltage of −100 mV. R308H currents were recorded using a holding voltage of −90 mV, steps from −100 mV to +180 mV in 40 mV increments and a tail voltage of −50 mV. Red dotted line denotes zero current. **f)** Gating currents recorded for Kv2.1 F412L in the presence of 100 mM external K$^+$. Holding voltage was −90 mV, steps were from −90 mV to +50 mV (10 mV increments) and a P/−4 protocol was used to subtract the leak and the capacitive currents. **g)** Q-V relations for the F412L mutant in 2- and 100-mM external K$^+$. Data for low K$^+$ are from Fig. 2d, e. Symbols represent mean, error bars represent S.E.M. and dark green solid curve is a fit of the Boltzmann Equation to data in high K$^+$ with $V_{1/2} = -24.8 \pm 0.5$ mV, $z = 3.7 \pm 0.3$ (n = 3 cells in 3 independent experiments).

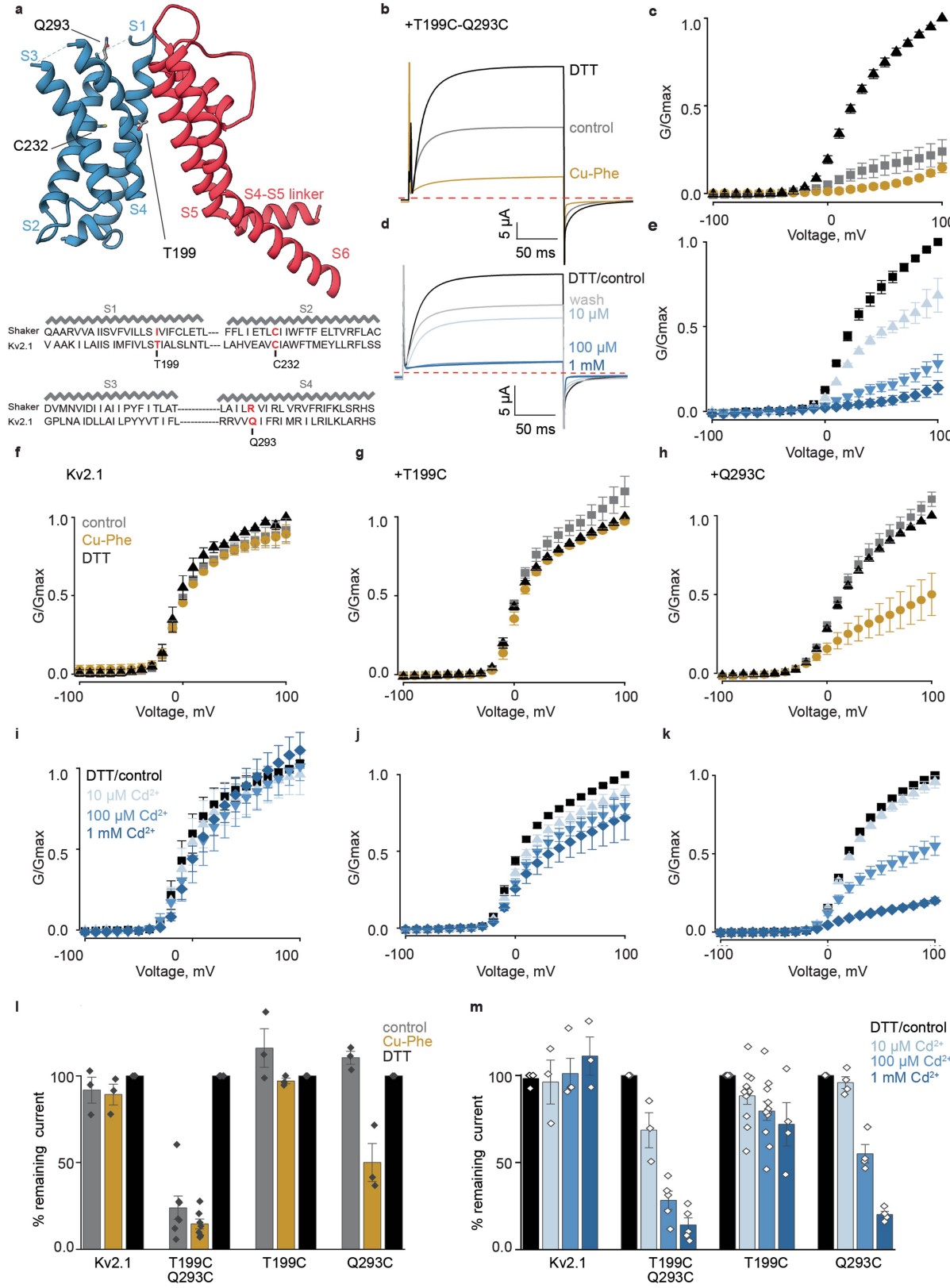

**Extended Data Fig. 3** | See next page for caption.

**Extended Data Fig. 3 | Gauging the movements of S4 in Kv2.1 (1-598) with disulfide and metal bridges. a)** Side view of the VSD of one subunit (blue) and the PD of the adjacent subunit (red). Q293 in the R1 position of S4, T199 in the S1 helix and C232 in the S2 helix are shown as sticks. Sequence alignment shows the TM segments within the VSDs of Shaker and Kv2.1. **b)** Superimposed traces obtained from the same oocyte expressing the T199C/Q293C double mutant of Kv2.1 treated as indicated. Holding voltage was −90 mV, test voltage was +100 mV, tail voltage was −50 mV and external K$^+$ was 50 mM. The control (gray) trace was obtained before any treatment, the Cu-Phe trace (dark yellow) in presence of Cu-Phe (1.5 μM–5 μM) and the DTT trace (black) after incubation with DTT (10 mM) for 10 min. **c)** G-V relations obtained for cells treated as in b using tail current measurements (-50 mV) and normalizing to the maximal current amplitude after DTT treatment (n = 7 cells in 2 independent experiments). **d)** Superimposed traces obtained from the same oocyte expressing the T199C/Q293C double mutant of Kv2.1 treated as indicated. Holding voltage was −90 mV, test voltage was +100 mV, tail voltage was −50 mV and external K$^+$ was 50 mM. The control (black) trace was obtained after incubation with DTT (10 mM) for 10 min before applying Cd$^{2+}$ at the indicated concentrations (blue traces) and then returning to control external solution (gray trace). **e)** G-V relations obtained for cells treated as in d using from tail current measurements (−50 mV) and normalized to the maximal current amplitude after DTT treatment (n = 5 cells in 2 independent experiments for control, 100 μM and 1 mM Cd$^{2+}$ and n = 3 cells in 2 independent experiments for 10 μM Cd$^{2+}$). **f)** G-V relations obtained for cells expressing WT Kv2.1 and treated as in b using tail current measurements (−50 mV) and normalizing to the maximal current amplitude after DTT treatment (n = 3 cells in 2 independent experiments). **g)** G-V relations obtained for cells expressing T199C and treated as in b using tail current measurements (n = 3 cells in 2 independent experiments). **h)** G-V relations obtained for cells expressing Kv2.1 Q239C and treated as in b using tail current measurements (n = 3 cells in 2 independent experiments). **i)** G-V relations obtained for cells expressing Kv2.1 WT and treated as in d using tail current measurements (−50 mV) and normalizing to the maximal current amplitude after DTT treatment (n = 4 cells in 2 independent experiments for control and 100 μM Cd$^{2+}$ and n = 3 cells in 2 independent experiments for both 10 μM and 1 mM Cd$^{2+}$). **j)** G-V relations obtained for cells expressing Kv2.1 T199C and treated as in d using tail current measurements (n = 11 cells in 4 independent experiments for control, 10 μM and 100 μM Cd$^{2+}$ and n = 4 cells in 4 independent experiments for 1 mM Cd$^{2+}$). **k)** G-V relations obtained for cells expressing Kv2.1 Q293C and treated as in d using tail current measurements (n = 4 in 2 independent experiments). **l)** Quantification of the effects of oxidizing (Cu-Phe) and reducing (DTT) conditions for a population of cells treated similarly to that in b. Currents were first elicited in control condition without any treatments and then after treated with Cu-Phe followed by DTT. Steady-state current at the end of test pulses to +100 mV were normalized to the maximal current amplitude after DTT treatment for the different constructs tested: Kv2.1 (n = 3 in 2 independent experiments), Kv2.1 T199C-Q239C (n = 7 in 2 independent experiments), Kv2.1 T199C (n = 3 in 2 independent experiments), Kv2.1 Q239C (n = 3 in 2 independent experiments). **m)** Quantification of the effect of Cd$^{2+}$ at the concentrations indicated by measuring the steady-state current at +100 mV and normalizing it by the current measured after DTT incubation for the different constructs tested: Kv2.1 (n = 4 cells in 2 independent experiments for control, n = 3 in 2 independent experiments for 10 μM Cd$^{2+}$, n = 4 in 2 independent experiments for 100 μM Cd$^{2+}$ and n = 3 cells in 2 independent experiments for 1 mM Cd$^{2+}$), Kv2.1 T199C-Q239C (n = 5 cells in 2 independent experiments for control, 100 μM and 1 mM Cd$^{2+}$ and n = 3 in 2 independent experiments for 10 μM Cd$^{2+}$), Kv2.1 T199C (n = 11 cells in 4 independent experiments for control, 10 μM and 100 μM Cd$^{2+}$, and n = 4 cells in 4 independent experiments for 1 mM Cd$^{2+}$), and Kv2.1 Q239C (n = 4 cells in 2 independent experiments for control and all Cd$^{2+}$ concentrations. In all panels error bars represent S.E.M. All experiments in this figure were with the structural construct: Kv2.1 (1-598).

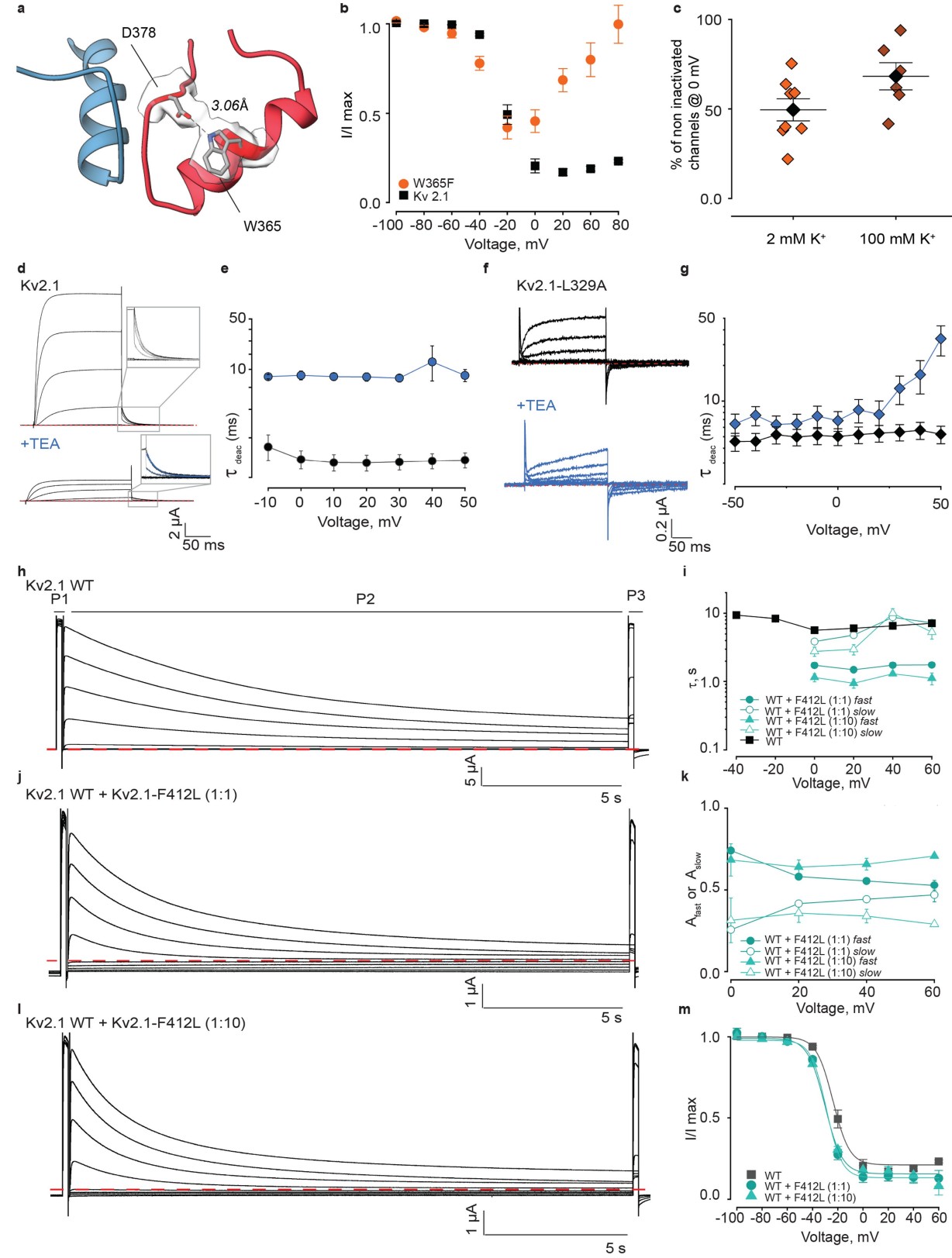

**Extended Data Fig. 4** | See next page for caption.

**Extended Data Fig. 4 | Exploring the non-conducting mechanism for Kv2.1 F412L. a)** Close up view of the model and EM map for D378 in the P-loop and W365 in the pore helix. Dashed line represents a hydrogen bond. **b)** Fraction of non-inactivated channels (P3/P1) in response to the P2 family of voltage steps from −100 to +60 mV for Kv2.1 (black squares; n = 3 in 2 independent experiments) and the W365F mutation (orange circles; n = 5 in 2 independent experiments). Similar protocol to that illustrated in Fig. 3d. External K⁺ was 2 mM. **c)** Fraction of non-inactivated channels for Kv2.1 W365F recorded in 2 or 100 mM external K⁺ using P/−4 subtraction. Coloured diamonds denote individual experiments (n = 8 cells in 2 independent experiments for 2 mM K⁺ and n = 6 in 2 independent experiments for 100 mM K⁺) and solid black diamonds represent mean. **d)** Family of current traces for Kv2.1 with 2 mM external K⁺ in the absence and presence of internal TEA. Test depolarizations were from −100 to +50 mV in 10 mV increments from a holding voltage of −90 mV using P/−4 subtraction. Insets show tail currents recorded at −50 mV. Red line indicates zero current. **e)** Plot of time constant of deactivation for Kv2.1 in the absence and presence of internal TEA (n = 3 in 2 independent experiments). **f)** Family of current traces for Kv2.1 L329A with 2 mM external K⁺ in the absence and presence of internal TEA. Test depolarizations were from −90 to +50 mV in 10 mV increments with holding and tail voltages of −90 mV using P/−4 subtraction. Red line indicates zero current. **g)** Plot of time constant of slow current measured for Kv2.1 L329A upon repolarization in the absence and presence of internal TEA (n = 7 in 4 independent experiments). **h)** Current families for Kv2.1 obtained using a three-pulse protocol with 2 mM external K⁺ and a holding voltage of −100 mV. The first pulse (P1) was to +60 mV, followed by a brief closure to −100 mV, the test pulse (P2) was from −100 to +60 mV for 20 s to allow the channels to inactivate, and a third pulse (P3) to the same voltage than P1 to assess the fraction of inactivated channels **i)** Plot of time constants (τ) of inactivation against P2 voltage. τ was obtained by fitting a single or double exponential function to the time course of the test current in P2. Data points are mean and error bars are S.E.M. for Kv2.1 (black squares; n = 3 in 2 independent experiments), Kv2.1+Kv2.1-F412L (1:1) (green circles; n = 3 in 2 independent experiments) and Kv2.1+Kv2.1-F412L (1:10) (green triangles; n = 4 in 2 independent experiments). **j)** Current families obtained as in h but when co-expressing Kv2.1 with the F412L mutant using a 1:1 molar ratio of cRNA. **k)** Amplitudes for the fast and slow components of the double-exponential fits obtained when co-expressing Kv2.1 with F412L. Same cells and n values as panel i. **l)** Current families obtained as in h but when co-expressing Kv2.1 with the F412L mutant using a 1:10 molar ratio of cRNA. **m)** Plot of fraction of non-inactivated channels during each P2 voltage step for Kv2.1 (black squares; n = 3 in 2 independent experiments), Kv2.1+Kv2.1-F412L (1:1) (green circles; n = 6 in 2 independent experiments) and Kv2.1+Kv2.1-F412L (1:10) (green triangles; n = 6 in 2 independent experiments) obtained by measuring the steady-state current at P3 normalized to P1. Smooth curves are fits of a Boltzmann equation (Kv2.1, $V_{1/2} = -23.3 \pm 1.0$ mV, z = 3.9 ± 0.3; Kv2.1:F412L 1:1, $V_{1/2} = -30.1 \pm 1.4$ mV, z = 4.1 ± 0.3; Kv2.1:F412L 1:10, $V_{1/2} = -30.1 \pm 2.1$ mV, z = 3.6 ± 0.3). For all panels error bars are S.E.M.

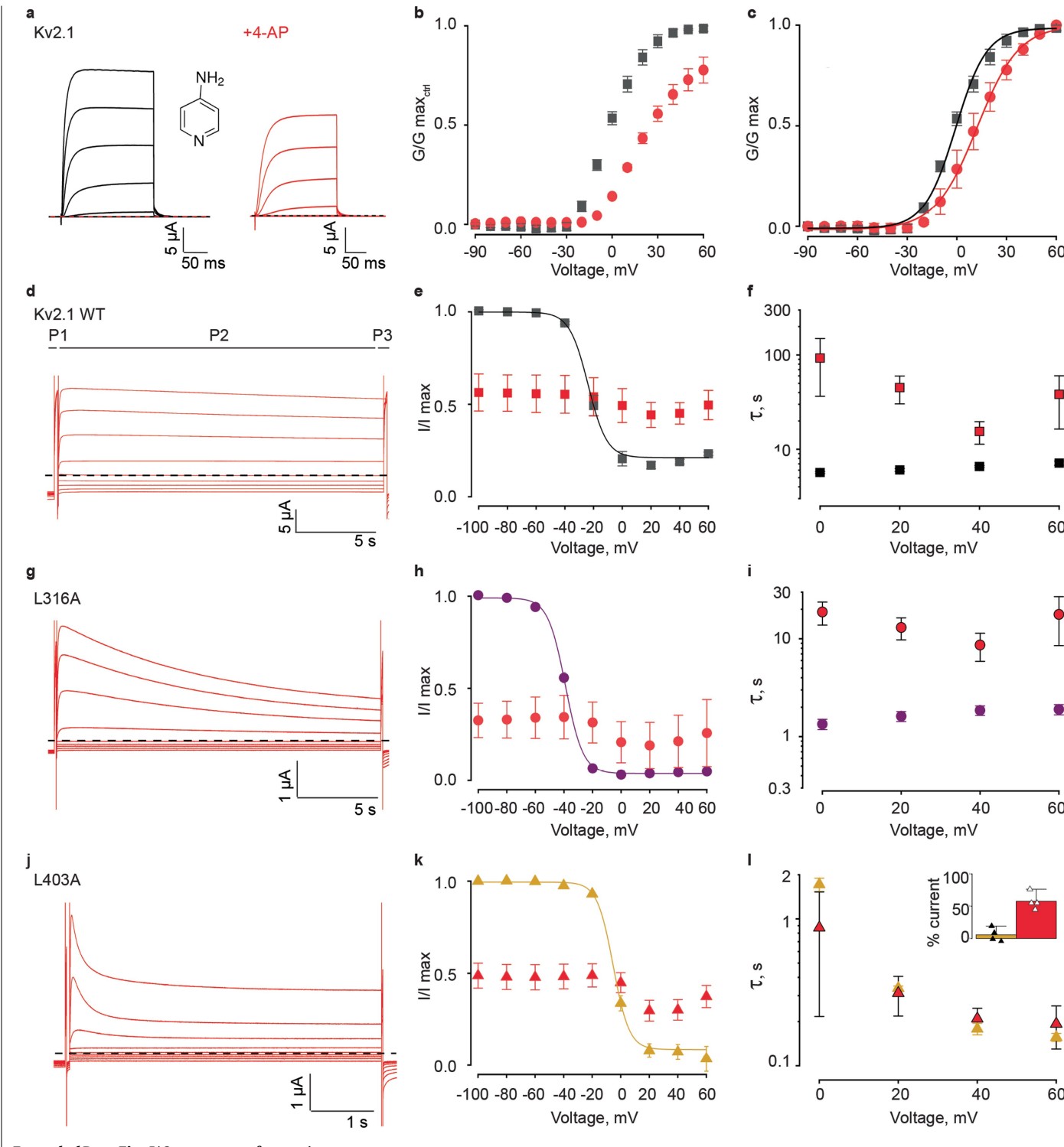

**Extended Data Fig. 5** | See next page for caption.

**Extended Data Fig. 5 | 4-AP interferes with inactivation in Kv2.1. a)** Current traces for Kv2.1 with 2 mM external K$^+$ in the absence and presence of 4-AP (10 mM). Test depolarizations were from −90 to +60 mV in 20 mV increments from a holding voltage of −90 mV using a tail voltage of −60 mV and P/−4 subtraction. Red and black dashed lines indicates zero current. **b)** G-V relations obtained measuring the peak tail currents from traces as shown in panel a for Kv2.1 (n = 3 in 2 independent experiments) before and after 4-AP addition. Tail currents were normalized to the maximum value in control solution. **c)** G-V relations independently normalized to the maximum tail current amplitude recorded in the absence or presence of 4-AP. Fits of the Boltzmann equation to the data are shown as solid curves. For control, $V_{1/2}$ = −1.1 ± 0.9 mV and z = 2.4 ± 0.1 For 4-AP, $V_{1/2}$ = 18.1 ± 0.9 mV and z = 2.1 ± 0.1. Same cells as in panel b. **d)** Current families for Kv2.1 obtained using a three-pulse protocol with 2 mM external K$^+$ in the presence of 4-AP (10 mM). Holding voltage of −100 mV, the first pulse (P1) was to +60 mV, followed by a brief closure to −100 mV, the test pulse (P2) was from −100 to +60 mV for 20 s to allow the channels to inactivate, and a third pulse (P3) to +60 mV to assess the fraction of inactivated channels. Black dashed line indicates zero current. See Fig. 3d for control traces in the absence of 4-AP. **e)** Plot of fraction of non-inactivated channels during the P2 voltage step (P3/P1) for Kv2.1 in control solution (black squares) and after application of 4-AP (red squares). Current amplitudes in 4-AP were normalized to that in control in the same cell. Data points are mean and n = 3 in 2 independent experiments. Solid line in control is a fit of a Boltzmann equation to the control data with $V_{1/2}$ = −23.3 ± 1.0 mV, z = 3.9 ± 0.3. **f)** Time constants for inactivation for Kv2.1 in control solution and in the presence of 4-AP (10 mM). Values for τ were obtained by fitting a single exponential function to the time course of current decay during P2 in panel d. Inactivation is barely detectable on this timescale in the presence of 4-AP and thus τ values in that condition are poorly defined.

Same cells as in panel e. **g)** Current families for Kv2.1 L316A obtained using a three-pulse protocol with 2 mM external K$^+$ in the presence of 4-AP (10 mM). Same protocol as in d. Dashed black line indicates zero current. See Fig. 3d for control traces in the absence of 4-AP. **h)** Plot of fraction of non-inactivated channels during the P2 voltage step (P3/P1) for Kv2.1 L316A in control solution (purple circles) and after application of 4-AP (red circles). Current amplitudes in 4-AP were normalized to that in control in the same cell. Data points are mean and n = 3 in 2 independent experiments. Solid line in control is a fit of a Boltzmann equation to the data with $V_{1/2}$ = −39.0 ± 0.5 mV and z = 4.3 ± 0.2. **i)** Time constants for inactivation for Kv2.1 L316A in control solution and in the presence of 4-AP (10 mM). Values for τ were obtained by fitting a single exponential function to the time course of current decay during P2 in panel g. Same cells as in panel h. **j)** Current families for Kv2.1 L403A obtained using a three-pulse protocol with 2 mM external K$^+$ in the presence of 4-AP (10 mM). Similar protocol to that in d except P1 and P3 were to +50 mV and P2 duration was 5 sec. Dashed black line indicates zero current. See Fig. 3d for control traces in the absence of 4-AP. **k)** Plot of fraction of non-inactivated channels during the P2 voltage step (P3/P1) for Kv2.1 L403A in control solution (yellow triangles) and after application of 4-AP (red triangles). Current amplitudes in 4-AP were scaled to that in control in the same cell. Data points are mean and n = 4 in 4 independent experiments. Solid line is a fit of a Boltzmann equation to the data with $V_{1/2}$ = −4.9 ± 0.8 mV and z = 4.3 ± 0.3. **l)** Time constants for inactivation for Kv2.1 L403A in control solution and in the presence of 4-AP (10 mM). Values for τ were obtained by fitting a single exponential function to the time course of current decay during P2. Same cells as in panel k. Bar graph insert represents the mean % of remaining current at the end of the trace measured before (yellow) and after 4-AP (red). Symbols represent individual experiments and error bars are S.E.M.

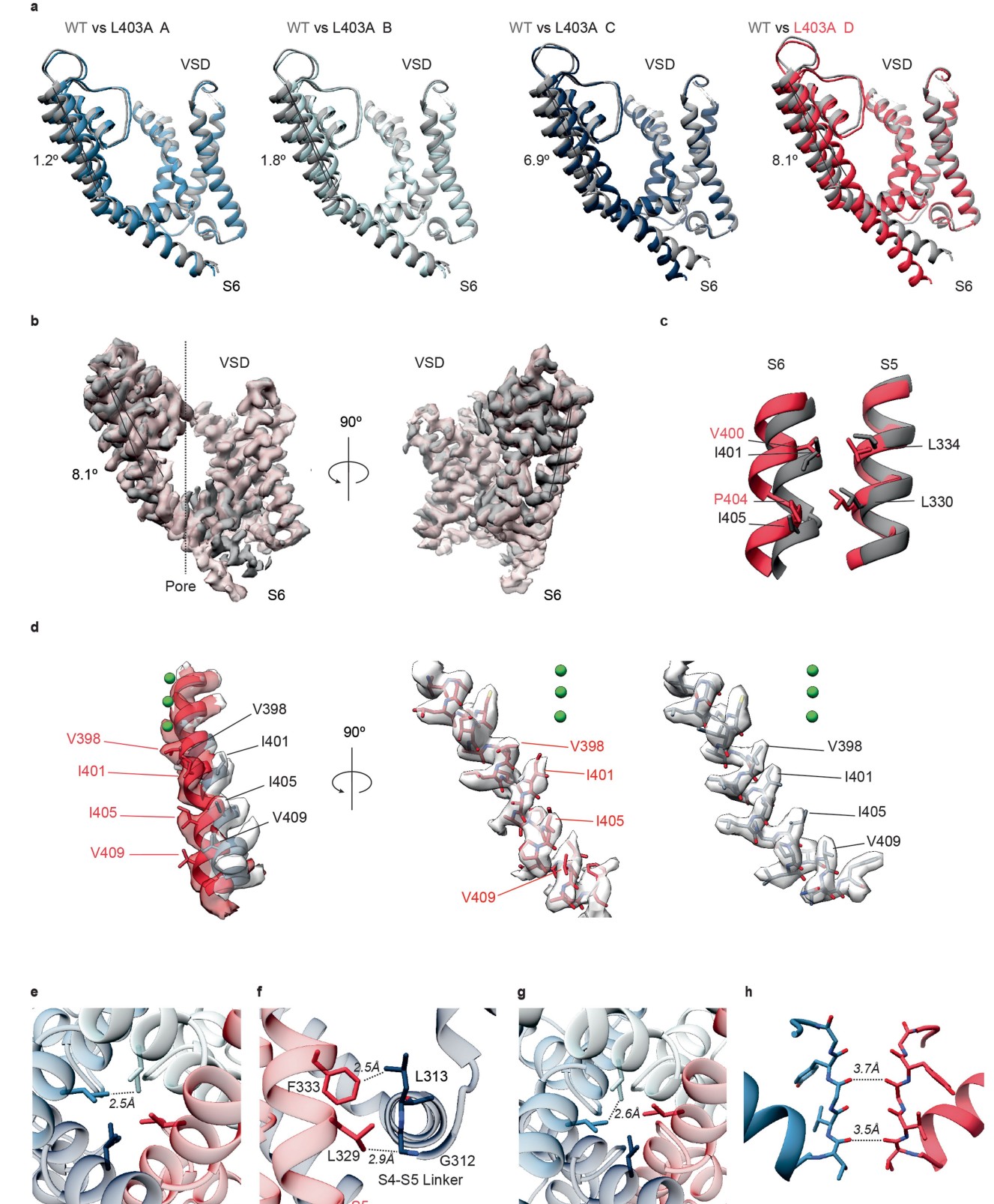

**Extended Data Fig. 6** | See next page for caption.

**Extended Data Fig. 6 | Structural comparison of Kv2.1 and the L403A mutant.**
**a)** Superimposition of Kv2.1 (gray) with each of the four protomers of the L403A mutant aligned using the VSDs. **b)** Cryo-EM map of a single subunit of Kv2.1 (gray) with the map for protomer D in the L403A mutant (red). **c)** Conformational changes at the interface between S5 and S6 helices between Kv2.1 (gray) and protomer D of the L403A mutant (red). **d)** Conformational changes in the S6 helix between Kv2.1 (gray) and protomer D of the L403A mutant (red). Cryo-EM maps for S6 helices from protomer D in the L403A mutant (red) and Kv2.1 (gray) are shown to the right. **e,f)** A symmetrical L403A tetramer generated by aligning four D protomers to Kv2.1 based on selectivity filter. Dash lines indicate clashes between neighboring I405 side chains (**e**), between the sidechain of L329 and backbone of G312 (**f**) and between the sidechain of F333 and that of L313 (**f**). **g,h)** A symmetrical L403A tetramer modeled by aligning four D protomers to Kv2.1 based on the VSDs. Dash lines show clashes between neighboring I405 side chains (**g**) and distances between backbone carbonyls within the selectivity filter that are shorter when compared to those of Kv2.1 and other K$^+$ channels whose filters are thought to be conducting (**h**).

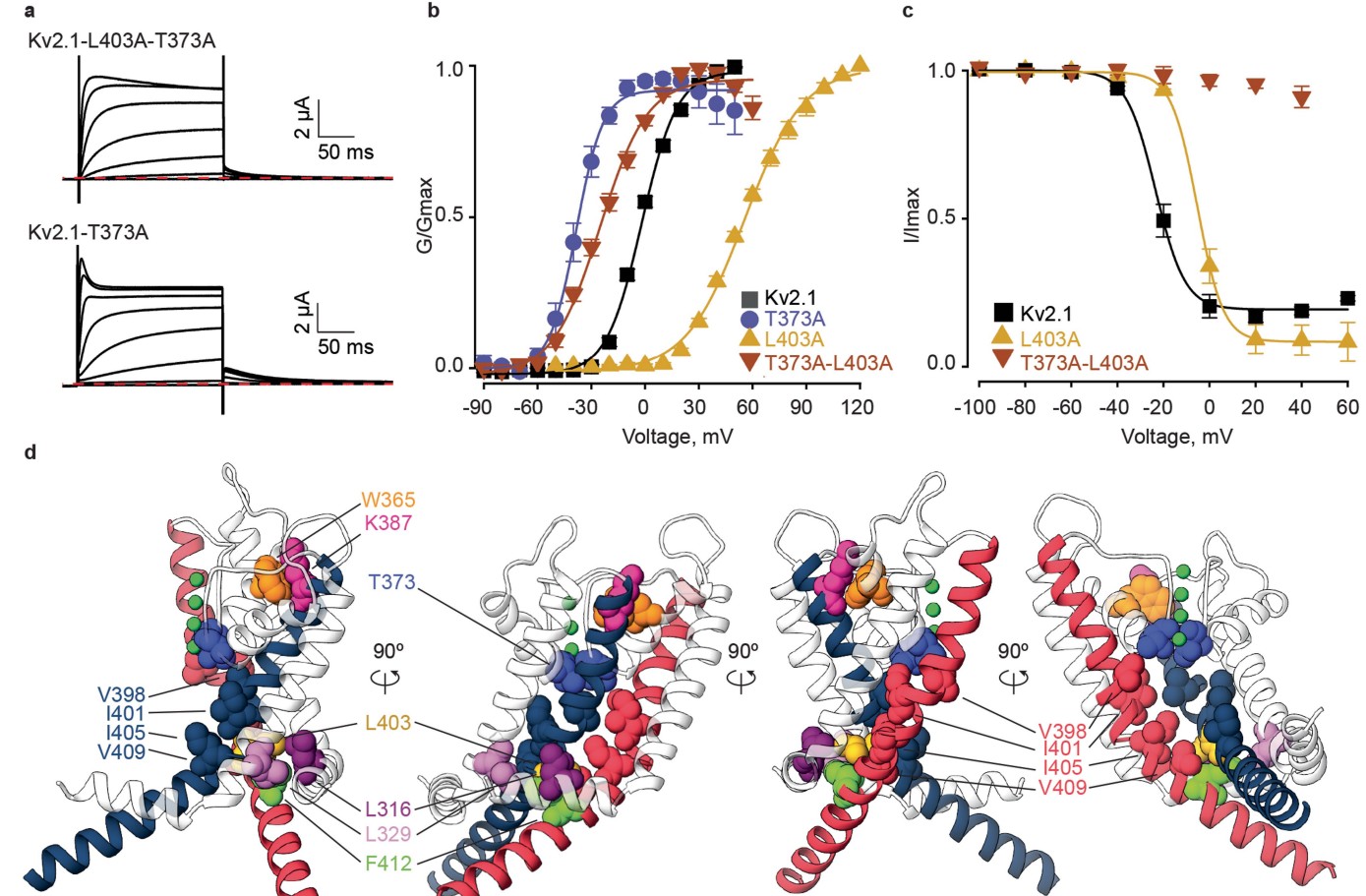

**Extended Data Fig. 7 | A selectivity filter mutant interferes with inactivation in Kv2.1 L403A. a)** Current traces for the Kv2.1-L403A-T373A double mutant and Kv2.1-T373A with 2 mM external K[+]. Test depolarizations for Kv2.1-L403A-T373A were from −90 to +60 mV in 10 mV increments from a holding voltage of −90 mV using a tail voltage of −60 mV with P/−4 subtraction. Kv2.1-T373A test depolarizations were from −90 to +50 mV in 10 mV increments from a holding voltage of −90 mV using a tail voltage of −70 mV. Red line indicates zero current. **b)** G-V relations obtained by measuring the peak tail currents from traces as shown in panel a for Kv2.1- T373A (blue circles, n = 6 in 4 independent experiments) and Kv2.1- L403A-T373A (inverted maroon triangles, n = 6 in 4 independent experiments). For comparison purposes, data for L403A and Kv2.1 are also shown from Fig. 3c. Symbols represent mean and solid line is a fit

of a Boltzmann equation to the data for Kv2.1-T373A ($V_{1/2}$ = −38.4 ± 1.5 mV and z = 3.4 ± 0.4), Kv2.1-T373A-L403A ($V_{1/2}$ = −25.1 ± 1.8 mV and z = 1.9 ± 0.1). **c)** Plot of fraction of non-inactivated during the P2 voltage step (P3/P1) using the same protocol as in Fig. 3d for Kv2.1-T373A-L403A (inverted maroon triangles, n = 4 in 2 independent experiments) in 2 mM external K[+]. Kv2.1 and Kv2.1-L403A are shown from comparison purposes (from Fig. 3f). **d)** Side views of the Kv2.1 model for two neighboring subunits, from the S4-S5 linker to the S6 (S6 coloured in dark blue or red). Residues tested in the present manuscript that modify inactivation are shown in different colours. Other residues that undergo rearrangements during inactivation as seen in the Kv2.1-L403A model (see Fig. 5, Extended Data Fig. 6) are shown in the same colour as the helix. For all panels error bars are S.E.M.

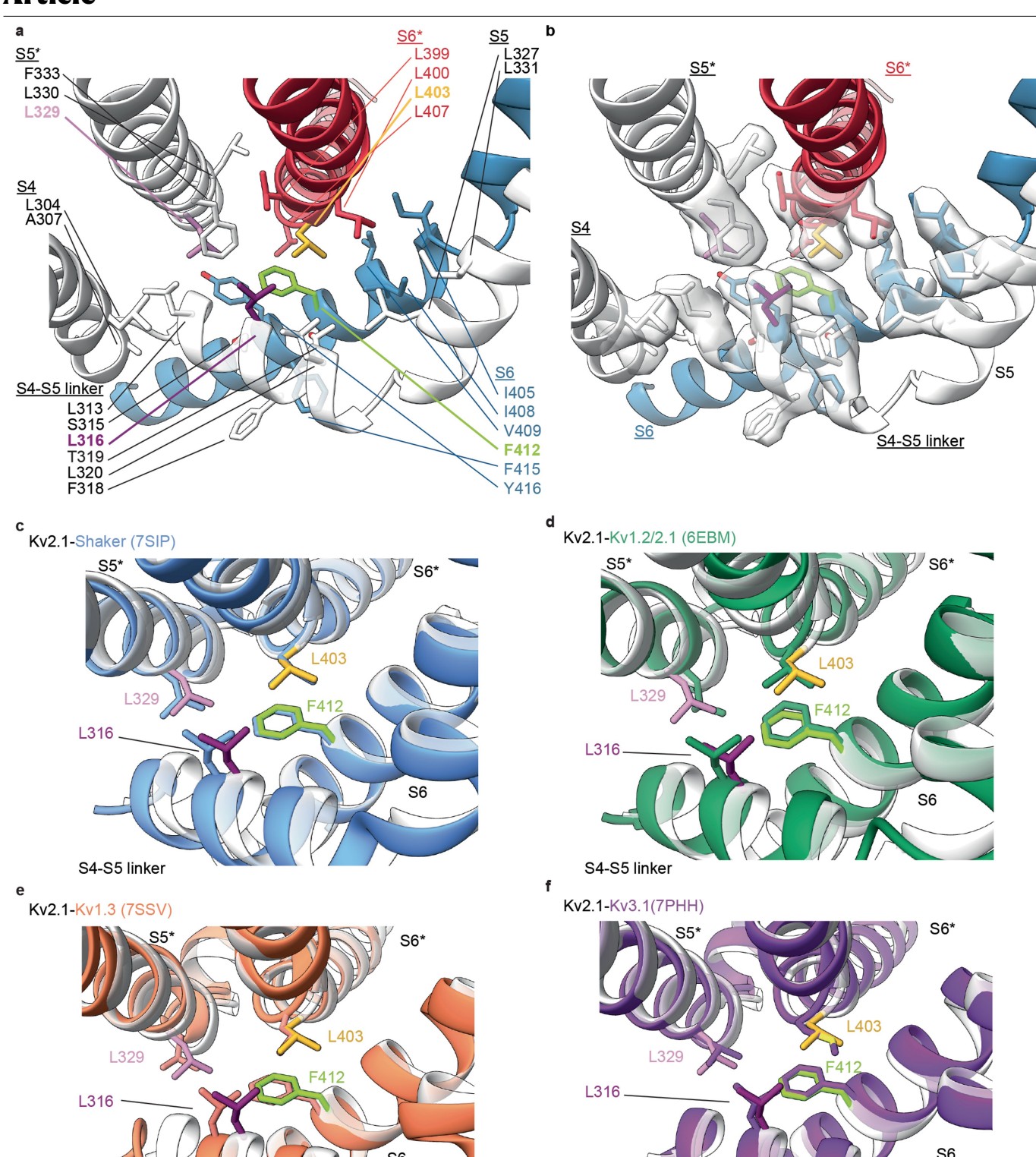

**Extended Data Fig. 8 | Additional hydrophobic residues interacting with the hydrophobic coupling nexus in Kv2.1 and structural alignment of Kv2.1 with other Kv channels. a)** View of the hydrophobic coupling nexus residues highlighted with the side chains depicted as sticks with F412 green, L316 purple, L329 light purple and L403 yellow, with additional hydrophobic residues depicted as sticks coloured based on the helix in which they are located. **b)** Same view and model as in panel a but also showing cryo-EM density for hydrophobic side chains. **c-f)** Close-up view of the hydrophobic coupling nexus residues highlighted with the side chains depicted as sticks with Kv2.1 residues labeled for Kv2.1 (white) and **c)** Shaker-IR (blue), **d)** Kv1.2/2.1 paddle chimera (green), **e)** Kv1.3 (orange) and **f)** Kv3.1 (purple).

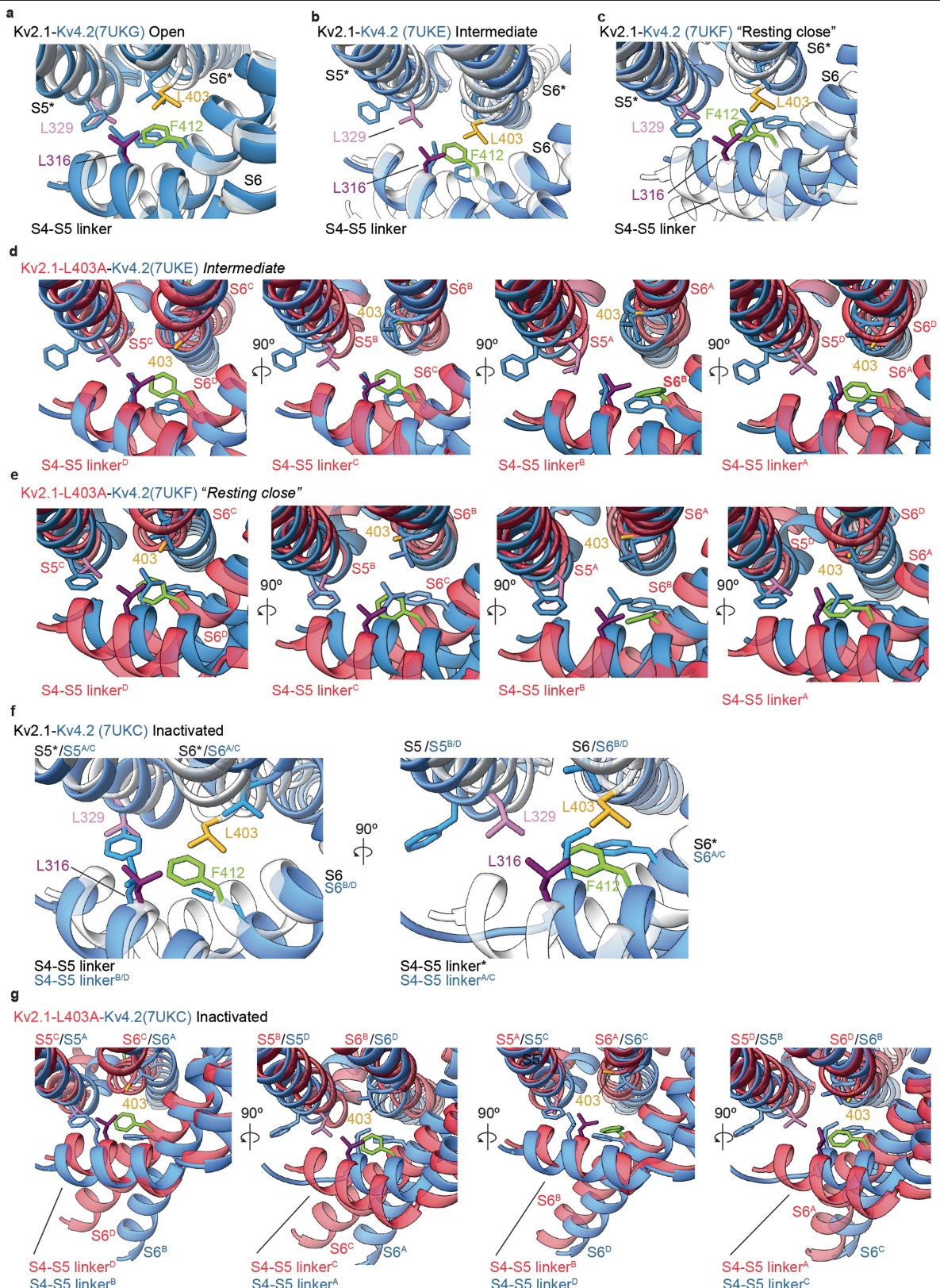

**Extended Data Fig. 9 | Structural alignment of Kv2.1 and Kv4.2 channels proposed to be in open, intermediate, resting/close or inactivated states.** **a)** Close-up view of the hydrophobic coupling nexus residues in Kv2.1 (white) and an open state of Kv4.2 (blue). **b)** Hydrophobic coupling nexus residues in Kv2.1 (white) and an intermediate state of Kv4.2 (blue). **c)** Hydrophobic coupling nexus residues in Kv2.1 (white) and a resting/close state of Kv4.2 (blue). **d)** Hydrophobic coupling nexus residues in the four protomers of the L403A mutant of Kv2.1 (red) and an intermediate state of Kv4.2 (blue). **e)** Hydrophobic coupling nexus residues in the four protomers of the L403A mutant of Kv2.1 (red) and a resting/close state of Kv4.2 (blue). **f)** Close-up view of the hydrophobic coupling nexus residues in Kv2.1 (white) and the two distinct protomers present in an inactivated state of Kv4.2 (blue). **g)** Hydrophobic coupling nexus residues in the four protomers of the L403A mutant of Kv2.1 (red) and the two distinct protomers present in an inactivated state of Kv4.2 (blue).

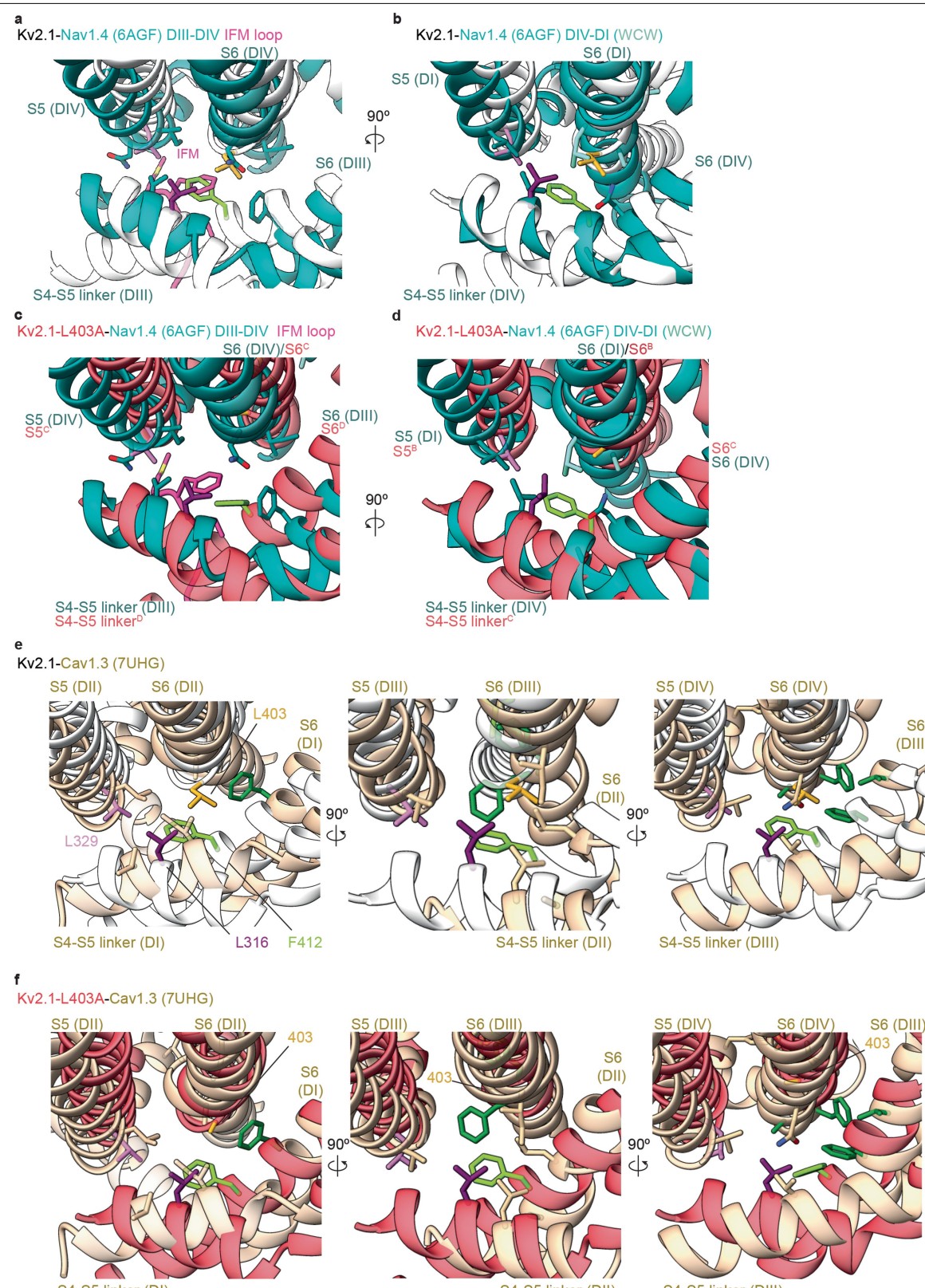

**Extended Data Fig. 10 | Structural alignment of Kv2.1 with Nav1.4 and Cav1.3 channels. a)** Close-up view of the hydrophobic coupling nexus residues in Kv2.1 (white) and between domains III and IV of Nav1.4 (blue). **b)** Hydrophobic coupling nexus residues in Kv2.1 (white) and between domains IV and I of Nav1.4 (blue). WCW mutants (see Discussion) are shown in stick representation (light blue). **c)** Hydrophobic coupling nexus residues in the L403A mutant of Kv2.1 (red) and between domains III and IV of Nav1.4 (blue). **d)** Hydrophobic coupling nexus residues in the L403A mutant of Kv2.1 (red) and between domains IV and I of Nav1.4 (blue). **e,f)** Close-up view of the hydrophobic coupling nexus residues highlighted with the side chains depicted as sticks with Kv2.1 residues labeled for Kv2.1 (white) and Cav1.3 (tan) **(e)** or Kv2.1-L403A (red) and Cav1.3 (tan) **(f)**. Dark green side chains identify residues where mutations enhance voltage dependent inactivation[59].

**Extended Data Table 1 | Cryo-EM data collection, refinement and validation statistics**

| | Kv2.1 WT (EMDB-40349) (PDB 8SD3) | Kv2.1 L403A (EMDB-40350) (PDB 8SDA) |
|---|---|---|
| **Data collection and processing** | | |
| Magnification | 130,000 | 105,000 |
| Voltage (kV) | 300 | 300 |
| Electron exposure (e–/Å$^2$) | 71 | 48 |
| Defocus range (μm) | -0.5 to -2.0 | -0.5 to -1.5 |
| Pixel size (Å) | 1.06 | 0.415 |
| Symmetry imposed | C4 | C1 |
| Initial particle images (no.) | 2,374,290 | 12,700,990 |
| Final particle images (no.) | 73,029 | 505,078 |
| Map resolution (Å) | 2.95 | 3.32 |
| FSC threshold | 0.143 | 0.143 |
| Map resolution range (Å) | 2.55-12.32 | 2.97-12.59 |
| | | |
| **Refinement** | | |
| Initial model used (PDB code) | 6EBM | 8SD3 |
| Model resolution (Å) | 3.3 | 3.4 |
| FSC threshold | 0.5 | 0.5 |
| Map sharpening $B$ factor (Å$^2$) | -95 | -100 |
| Model composition | | |
| Non-hydrogen atoms | 7832 | 7414 |
| Protein residues | 948 | 939 |
| Ligands | 28 | 19 |
| $B$ factors (Å$^2$) | | |
| Protein | 67.22 | 75.63 |
| Ligand | 20.08 | 20.69 |
| R.m.s. deviations | | |
| Bond lengths (Å) | 0.009 | 0.006 |
| Bond angles (°) | 0.725 | 0.737 |
| Validation | | |
| MolProbity score | 1.47 | 1.79 |
| Clashscore | 8.81 | 14.13 |
| Poor rotamers (%) | 0 | 0 |
| Ramachandran plot | | |
| Favored (%) | 98.7 | 97.27 |
| Allowed (%) | 1.3 | 2.73 |
| Disallowed (%) | 0 | 0 |

# Reporting Summary

## Statistics

For all statistical analyses, confirm that the following items are present in the figure legend, table legend, main text, or Methods section.

| n/a | Confirmed | |
|---|---|---|
| ☐ | ☒ | The exact sample size (*n*) for each experimental group/condition, given as a discrete number and unit of measurement |
| ☐ | ☒ | A statement on whether measurements were taken from distinct samples or whether the same sample was measured repeatedly |
| ☒ | ☐ | The statistical test(s) used AND whether they are one- or two-sided *Only common tests should be described solely by name; describe more complex techniques in the Methods section.* |
| ☒ | ☐ | A description of all covariates tested |
| ☐ | ☒ | A description of any assumptions or corrections, such as tests of normality and adjustment for multiple comparisons |
| ☐ | ☒ | A full description of the statistical parameters including central tendency (e.g. means) or other basic estimates (e.g. regression coefficient) AND variation (e.g. standard deviation) or associated estimates of uncertainty (e.g. confidence intervals) |
| ☒ | ☐ | For null hypothesis testing, the test statistic (e.g. *F*, *t*, *r*) with confidence intervals, effect sizes, degrees of freedom and *P* value noted *Give P values as exact values whenever suitable.* |
| ☒ | ☐ | For Bayesian analysis, information on the choice of priors and Markov chain Monte Carlo settings |
| ☒ | ☐ | For hierarchical and complex designs, identification of the appropriate level for tests and full reporting of outcomes |
| ☒ | ☐ | Estimates of effect sizes (e.g. Cohen's *d*, Pearson's *r*), indicating how they were calculated |

*Our web collection on statistics for biologists contains articles on many of the points above.*

## Software and code

Policy information about availability of computer code

| Data collection | Leginon 3.6 and SerialEM 3.8.1 for cryo-EM and pClamp 10.7 for electrophysiology |
|---|---|
| Data analysis | RELION (v3.0 and 4.0), MotionCor2, CTFFIND4, Gautomatch (0.56), Phenix 1.19.1, Coot 0.9.8.1, PyMol 2.4.1 and Chimera 1.15, MDAnalysis 2.4.0 and HOLE 2.2.005 for cryo-EM and pClamp 10.7 and OriginLab 2020 for electrophysiology |

For manuscripts utilizing custom algorithms or software that are central to the research but not yet described in published literature, software must be made available to editors and reviewers. We strongly encourage code deposition in a community repository (e.g. GitHub). See the Nature Portfolio guidelines for submitting code & software for further information.

## Data

Policy information about availability of data

All manuscripts must include a data availability statement. This statement should provide the following information, where applicable:
- Accession codes, unique identifiers, or web links for publicly available datasets
- A description of any restrictions on data availability
- For clinical datasets or third party data, please ensure that the statement adheres to our policy

All data needed to evaluate the conclusions in the paper are present in the paper, Extended Data Figures or the Supplementary Materials. Maps of Kv2.1 and the L403A mutant have been deposited in the Electron Microscopy Data Bank (EMDB) under accession codes EMD-40349 and EMD-40350, respectively. Models of

# Research involving human participants, their data, or biological material

Policy information about studies with human participants or human data. See also policy information about sex, gender (identity/presentation), and sexual orientation and race, ethnicity and racism.

| | |
|---|---|
| Reporting on sex and gender | n/a |
| Reporting on race, ethnicity, or other socially relevant groupings | n/a |
| Population characteristics | n/a |
| Recruitment | n/a |
| Ethics oversight | n/a |

Note that full information on the approval of the study protocol must also be provided in the manuscript.

# Field-specific reporting

Please select the one below that is the best fit for your research. If you are not sure, read the appropriate sections before making your selection.

☒ Life sciences  ☐ Behavioural & social sciences  ☐ Ecological, evolutionary & environmental sciences

For a reference copy of the document with all sections, see nature.com/documents/nr-reporting-summary-flat.pdf

# Life sciences study design

All studies must disclose on these points even when the disclosure is negative.

| | |
|---|---|
| Sample size | Statistical methods were not used to determine sample size. Sample size for cryo-EM studies was determined by availability of microscope time and to ensure we obtain sufficient resolution for model building. Sample size for electrophysiological studies was determined empirically by comparing individual measurements with population data obtained under differing conditions until convincing differences or lack thereof were evident. |
| Data exclusions | For electrophysiological experiments, exploratory experiments were undertaken with varying ionic conditions and voltage-clamp protocols to define ideal conditions for measurements reported in this study. Although these preliminary experiments are consistent with the results we report, they were not included in our analysis due to varying experimental conditions. Once ideal conditions were identified, electrophysiological data were collected for control and mutant constructs until convincing trends in population datasets were obtained. Individual cells were also excluded if cells exhibited excessive initial leak currents at the holding voltage (>0.5 μA), if currents arising from expressed channels were too small (<0.5 μA), making it difficult to distinguish the activity of expressed channels from endogenous channels, or if currents arising from expressed channels were too large, resulting in substantial voltage errors or changes in the concentration of ions in either intracellular or extracellular solutions. |
| Replication | Information on sample size is provided in figure legends throughout the manuscript. |
| Randomization | Randomization was not used in this study. The effects of different conditions or mutations on Kv2.1 channels heterologously expressed in individual cells was either unambiguously robust or clearly indistinguishable from control conditions. |
| Blinding | Blinding was not used in this study. The effects of different conditions or mutations on Kv2.1 channels heterologously expressed in individual cells was either unambiguously robust or clearly indistinguishable from control conditions. |

# Reporting for specific materials, systems and methods

We require information from authors about some types of materials, experimental systems and methods used in many studies. Here, indicate whether each material, system or method listed is relevant to your study. If you are not sure if a list item applies to your research, read the appropriate section before selecting a response.

## Materials & experimental systems

| n/a | Involved in the study |
|-----|----------------------|
| ☒ ☐ | Antibodies |
| ☐ ☒ | Eukaryotic cell lines |
| ☒ ☐ | Palaeontology and archaeology |
| ☐ ☒ | Animals and other organisms |
| ☒ ☐ | Clinical data |
| ☒ ☐ | Dual use research of concern |
| ☒ ☐ | Plants |

## Methods

| n/a | Involved in the study |
|-----|----------------------|
| ☒ ☐ | ChIP-seq |
| ☒ ☐ | Flow cytometry |
| ☒ ☐ | MRI-based neuroimaging |

## Eukaryotic cell lines

Policy information about cell lines and Sex and Gender in Research

| | |
|---|---|
| Cell line source(s) | Sf9 and tsA201 cells were originally obtained from Thermo Fischer and Sigma-Aldrich, respectively. |
| Authentication | Cell lines used were not authenticated. |
| Mycoplasma contamination | Mycoplasma contamination was tested and found to be negative |
| Commonly misidentified lines (See ICLAC register) | commonly misidentified cell lines were not used in this study |

## Animals and other research organisms

Policy information about studies involving animals; ARRIVE guidelines recommended for reporting animal research, and Sex and Gender in Research

| | |
|---|---|
| Laboratory animals | female Xenopus laevis frogs, 1-2 years of age, obtained from Xenopus I |
| Wild animals | no wild animals were used in the study |
| Reporting on sex | Female |
| Field-collected samples | no field collected samples were used in the study |
| Ethics oversight | The animal care and experimental procedures were performed in accordance with the Guide for the Care and Use of Laboratory Animals and were approved by the Animal Care and Use Committee of the National Institute of Neurological Disorders and Stroke (animal protocol number 1253). |

Note that full information on the approval of the study protocol must also be provided in the manuscript.

