## [Peer Review File · Nature]

Manuscript Title: Inactivation of the Kv2.1 channel through electromechanical coupling

Reviewer Comments & Author Rebuttals

Reviewer Reports on the Initial Version:

Referees' comments:

Referee #1 (Remarks to the Author):

A. Summary of the key results

Fernández-Mariño and colleagues have identified a novel inactivation motif in Kv2.1 channels. This site, comprised of four nonpolar residues coming together at the intracellular end of S6, appears to be distinct from those involved in traditional N-type and C-type inactivation processes. The identification was achieved through the combined use of electrophysiological functional assays and cryo-EM structural methods, applied to both the wild-type and an inactivation mutant embedded in nano-discs.

B. Originality and significance

The methodologies and approaches utilized in this study are well-established and the discovery of the so-called "hydrophobic nexus" is notable. I concur with the authors' assertions regarding the potential ramifications of this study. Analogous mechanisms and motifs may exist in other voltage-gated ion channels, including voltage-gated sodium and calcium channels. The intimate relationship between the hydrophobic component and the S4-S5 linker may also suggest how certain voltage-gated ion channels can recover from inactivation without needing to reopen. If this proves correct, it will represent a significant advancement in our understanding.

While some issues remain unresolved, the core messages intended by this study are quite noteworthy. Ideally, the authors would have elucidated the F412L structure such that the correlation between function and structure would have been firmer. However, even in the absence of the F412L structure or the homomeric L403A structure (bearing in mind the potential uncertainty regarding whether the structure truly depicts a stable inactivated state corresponding to functional measurements), the overall messages appear convincing and interesting enough. I am quite intrigued by the proposal that the IFM segment of voltage-gated Na⁺ channels impairs the nexus interaction and allows the inactivation process to occur.

C. Data & methodology: validity of approach, quality of data, quality of presentation

The approach and the data quality are fine. They may be a degree inconsistency regarding which experimental conditions are used (data traces with 2 mM K⁺ and other concentrations); the authors could better justify those choices made. Overall, the approach is straightforward and I do not see any concerns.

D. Appropriate use of statistics and treatment of uncertainties

I see no issues here. I appreciated that the authors refrained from null hypothesis testing. It would

be beneficial if they showed all data points in addition to the mean and SEM values. For example, there are multiple bar graphs where the authors can easily superimpose all individual data points. Also, the authors could probably state what, for example $n = 3$ means – 3 eggs from one donor/one RNA injection, etc.

E. Conclusions: robustness, validity, reliability

The overall messages are strong. However, the terminology the authors coined – “hydrophobic coupling nexus” – might need reconsideration. Please see below for more.

F. Suggested improvements (and questions)

Specific suggestions listed. Some are really minor. Some are subjective personal bias issues.

- “Hydrophobic coupling nexus”

From the mutagenesis results, it is evident that select nonpolar residues, namely F412, L316, L329, and L403, play crucial roles. Both F and L are typically considered to be nonpolar and perhaps hydrophobic. The term “hydrophobic coupling nexus” implies (and the authors also suggest?) that hydrophobic interactions among them are indeed essential to prevent inactivation. In this context, the more complete inactivation phenotype in the mutant F412L seems puzzling to me if I just consider only hydrophobic interactions. The electrons in the F aromatic side chain are more delocalized than the electrons in the L aliphatic isobutyl side chain; the L side chain should be more hydrophobic? In my review of the structure files provided, the F412 side chain is stabilized by the Y416 side chain in the wild-type channel and also in 3 of the 4 subunits in the L304 channel – do I have this right?. These F412-Y416 interactions are not totally straight-on face-to-face interactions but they are slightly off-centered staggered interactions. My understanding is that such slightly staggered interactions are quite common. In the mutant protomer D, this arrangement is notably impaired. Thus at the end, it may not be the hydrophobic interactions that are important for keeping the channel away from inactivation but it may turn out to be this pi-pi interaction between F412 and Y416. As a result, “hydrophobic coupling nexus” might not be the best term. Something like “internal coupling nexus”? As presented in this manuscript, the exact mechanisms are not that clear.

In addition, the above observations, if correct, suggest that some mutations encompassing F412 and Y416 may be in order – at least, I would like to see them (and I expect that many readers would also).

- “allosteric mechanisms of inactivation”...

The term “allosteric” is so abused and misused. It would be much if the authors could avoid this (unless this term is specifically defined and used in the manuscript).

- Page 4 “The structure of Kv2.1 ... are conserved”. This sentence seems a bit odd to me. The structure reveals that the residues are conserved?

- Fig. 4A. Perhaps, highlight R308 in the figure also?

- Page 4 “... that predominates at 0 mV...most of the charges accessible to the external solution”. Some readers may be confused base on the GV data in Fig. 1A. The relative G value there is only 0.5.

Also it may be helpful to remind the readers about the previous estimates of the number of equivalent gating charges involved in Kv2.1.

- Fig. 2B Data sweeps. Some readers may skip reading the legend. It may be worthwhile to point out the data sweeps are recorded with 100 mM K⁺ outside.
- Fig. 2D F412L data sweeps. Some may want to see data sweeps with a higher concentration of K⁺ (especially in comparison with C-type inactivation).
- Page 6 “Our proposed mechanism of inactivation in Kv2.1 ...” This seems somewhat out of place to me. Exactly what the inactivation mechanism is has not been fully explained prior to this sentence.
- W365F appears to activate more slowly also. Any guess as to why?
- Fig. 4B time scale bar labels. Among the four sweep sets, the Shaker-W434F sweeps (top left) look the fastest. But Fig. 4C says that only the W434F+TEA sweeps look slower. I am confused.
- The L403A structure. I would like to know (for my own education) why the wild-type structure at the end was refined with C4 symmetry and the L403 structure at the end was made with C1 symmetry. The C1 symmetry constrained worked fine for L403, revealing that one of the subunits had a different conformation. How did the decision to use the C1 symmetry option come about?
- Page 30 “... in 150 mL recording chambers...”. Is this really correct?
- Extended data Fig. 3. It may be useful remind what the cylinders and colors are in A and state what the three curves in B (masked, not masked, etc?).
- Extended data Fig. 5. Did you see any use dependence / holding voltage dependence for Cu-Phe and Cd? It may be good to say how often pulses were applied.
- Extended data Fig. 5 GV curves – smaller and shallower components at depolarized voltages. What are those components? Such components are much less noticeable in Fig. 1.
- Extended data Fig. 6. The WT data traces in Fig. 1 were recorded with 2 mM K⁺ outside. If the mutant sweeps with 2 mM K⁺ outside are presented, comparison with the WT data would be easier (I am not a big fan of using 2 mM K⁺ outside with oocytes, however).
- Extended data Fig. 7B. The voltage dependence reminds me of “U-type” inactivation reported for Kv1.5.
- Extended data Fig. 9. Given what is shown in A, I think it would be good to make all the data traces recorded with 4-AP red (D, G, J).
- Extended data Fig. 12B. Please state what the curves are.

G. References: appropriate credit to previous work?

The referencing appears adequate (although the breadth of the field may leave room for differing opinions). There are a few formatting issues that need to be cleaned up at a later stage.

H. Clarity and context: lucidity of abstract/summary, appropriateness of abstract, introduction and conclusions.

Please see my comments about the terms “hydrophobic coupling nexus” and “allosteric mechanisms”. The final conclusions and implications sections might benefit from slight truncation.

Referee #2 (Remarks to the Author):

The manuscript “Structures of the Kv2.1 channel and mechanism of inactivation through electromechanical coupling” by Fernández-Mariño and colleagues describes the novel structure of the wild-type Kv2.1 voltage-activated potassium channel; emphasizing the results description and discussion on the transmembrane domain, since the cytoplasmic domain was not resolved. By mapping mutations causing epileptic encephalopathies in the new channel structure, the authors spotted a region that couples the voltage sensor with the pore. This mechanism seems to be distinct to what it has been described for the Shaker Kv channel.

I found their results very interesting, especially because the authors demonstrate that Kv2.1 and Shaker have commonalities and differences. For instance, these channels S4 helices move similar distances, as determined by electrophysiology and disulfide bonds, but the C-type inactivation described for Shaker does not apply to Kv2.1, as determined by electrophysiology. In the spirit of this comment, I would suggest toning down the discussion that compares Kv2.1 with sodium and calcium channels. The reason for this is twofold: 1) Not all the structures that the authors compared with were obtained in lipid nanodiscs. Hence the absence or presence of lipids could favor different helices conformations. 2) There are no experimental results on this manuscript to support that calcium or sodium channels undergo a similar electromechanical coupling.

In the introduction section:

- Rewrite clarify this sentence in the introduction: “In response to sustained membrane depolarization, Kv channels inactivate, decreasing the flow of ions and critically influencing their contributions to electrical signaling constituting a form of short-term memory”.

In the results section:

- What is the size of the FL protein?
- Extended Figure 1 should show which regions of the protein were cut for purification
- Define: VAPA/B 38-40 and MSP 1E3D1.
- I would suggest quantifying excellent: “and the maps were relatively uniform and high quality throughout, with excellent density”
- Extended data figure 3E seems to be cropped.
- Try to be more quantitative in your descriptions, for example in the following sentence define what well-resolved means “some of which are particularly well-resolved (Fig. 1C,D)”.

- Make sure that the figures are cited in order.
- Add previously identified in a prokaryotic K channel in this sentence: “There are also strong EM densities at four positions with the filter along the central axis (Fig. 1E), suggesting that the filter is occupied by ions at the four sites previously identified 46”.
- This sentence needs to be rewritten: “Although each of these epileptic encephalopathy mutations are interesting, illustrating how disease can result from both loss and gain of function in Kv2.1, we were inspired to study the F412L mutant in greater detail because its non-conducting phenotype is reminiscent of a mutation in the Shaker Kv channel 53,54 that enabled the mechanism of slow C-type inactivation to be elucidated 13,27,28.”
- What does optimally mean in the following sentence? Is this in comparison to other channels? “the hydrophobic interactions within the nexus involving F412 appear to be optimally positioned for movements of...”
- The cartoons on figure 6 do not help highlighting the difference between the wt and the mutant structure. Figure 5I-J does a great job because the differences are striking. May be there is no need to have a cartoon, and rather use the space to move some of the many extended figures to the main manuscript.

Referee #3 (Remarks to the Author):

The manuscript by Fernandez-Marino et al. describes a comprehensive study of the rat Kv2.1 voltage-gated potassium channel which plays an important role in the central nervous system and mutations of the human Kv2.1 channel have been associated with epileptic encephalopathy. The authors utilize a combination of cryo-EM and electrophysiology to characterize a C-terminal truncated, but fully functional Kv2.1 variant and a range of disease mutations which lead to the identification of interesting phenotypes with accelerated inactivation for several mutants located within a hydrophobic nexus formed between neighboring S6 helices and the S4/S5 linker. This region is known to be important for electromechanical coupling and conserved in most domain-swapped voltage-gated cation channels. Extensive electrophysiological characterization (including gating current measurements, Cys engineering for disulfide/Cd²⁺ Hg²⁺ trapping to manipulate S4 movements, as well as measurements in the presence of TEA and 4-AP to sense the status of the internal pore-flanking S6 gate) is carried out to characterize F412L and other Kv2.1 disease mutants located in the hydrophobic nexus region. It is particularly striking that the F412L mutant can be rescued from being nonconductive by adding 4-AP –which counteracts inactivation by slowing the closure of the lower S6 gate. Furthermore, it is shown that the W365F mutation of Kv2.1 corresponding to the well-studied W434F shaker mutant remains conducting and inactivates more slowly than wild-type – a phenotype which is dramatically different from the shaker W434F phenotype with accelerated inactivation. Together, these functional results provide strong support for the interpretation that the nexus region plays a key role for the mechanism of inactivation of

Kv2.1 channels which is distinct from classical C-type activation of Shaker channels and involves dynamic changes in the electromechanical coupling rather than changes in the selectivity filter. The probably most compelling evidence for this new model stems from a cryo-EM structure of the L403A mutant within the nexus of Kv2.1 which exhibits an interesting asymmetrical channel conformation with two subunits having distinct changes in S6 compared to wild-type. This results in removal of the kink in the PXP region (characteristic for the open conformation) and causes obstruction of the pore in the lower gate region, hence explaining the fast inactivating phenotype of this variant. The structure of the selectivity filter and voltage sensor domain are largely unaltered by the mutation. In the discussion, the authors present useful comparisons to the respective regions in Kv4 channels and to Cav/Nav channels. It is suggested that the N-type inactivation of sodium channels is rather an allosteric effect of the IFM motif which inserts itself into the equivalent region of the hydrophobic coupling nexus in Nav1.4 channels (as opposed to the previously proposed mechanism which assumed that the IFM motif directly blocks the pore) and the potential therapeutic relevance for small molecule drugs modulating inactivation kinetics is highlighted.

All in all, the presented work is expertly done and represents a thorough investigation of a medically highly important ion channel and these novel findings may have further implications for the mechanistic understanding of the broader family of voltage-gated cation channels.

Electrophysiological experiments are carried out with sufficient n for meaningful statistical analysis and all data points have error bars representing S.E.M which is appropriate for the studies. Accuracy of the fit for the curves is also indicated for all biophysical parameters determined from the electrophysiological data. The cryo-EM maps and structural models are of sufficient quality to support the interpretation of the structural data. The study is well-written and the conclusions are clear and well supported by the experimental results. Therefore, the manuscript is suitable for publication in Nature and I have only a few minor points that should be addressed prior to acceptance of the manuscript:

Minor points:

- 1.) Please add a scale bar in the cryo-EM micrographs shown in Extended Figures 2 and 11
- 2.) Page 4, top paragraph: "captured in an active state that predominates at 0 mV" -> looks more like 50/50 between open/closed, since the midpoint potentials shown in Figure 1B are close to 0 mV (0.8 for full length Kv2.1 and 1.3 mV for the truncated EM construct)
- 3.) Page 5 and Fig 1F Ext Figure 4 C "...shows key basic residues that have been found to influence the single channel conductance" -> no individual residues shown, just a surface colored according to electrostatic potential. Adjust the Figure or change "residues" to "region"
- 4.) Page 7, middle: "In the case of L403, the extent..." -> L403A
- 5.) Page 7 and Figure 4 D, E, Ext. Fig 8: The analysis of heteromeric channels formed between wt and F412L mutant is interesting. It is however not mentioned whether the disease mutation is heterozygous and exhibits a dominant negative phenotype? This would make sense. Include a reference if this has been published.
- 6.) Figure 6 legend: "Cartoons representing the open state and a symmetric inactivated state" -> why is a symmetric inactive state shown if the structure of L403A has an asymmetrical arrangement of the four subunits and the authors state that there would be clashes between subunits if a symmetrical channel would be created by superposition of the "inactive" C and D subunits on a and B (Extended Fig 14)? The authors suggest that small rearrangements may reduce these clashes and the channel could relax into a symmetrical state with 4 inactive-conformation protomers without

clashes, but without results from a coarse-grained MD simulation supporting this idea or experimental data, a 4-fold symmetrical inactive state is purely hypothetical.

7.)Method section: please add the concentration of Kv2.1 nanodisc sample used for making cryo-EM grids. It currently states only “concentrated samples of Kv2.1 or the L403A mutant in nanodiscs (3 μ L) were applied to glow-discharged Quantifoil grid”

8.)Extended Figure legends are sometimes a bit short and would benefit from more information, eg. Extended Fig. 3 D and Figure 12 B show FSC curves (from Relion?) but there is no legend for the three traces in red, green and blue. I suppose this is masked (bold black) and green is unmasked and red is phase randomization.

9.)Extended Figure 16: marking different channel regions on top of the sequence alignment (eg. S4/S5 linker, VSD, S6 etc) would make this figure easier to understand.

10.)While drafting this report, a preprint from Fred Sigworth and coworkers appeared in bioRxiv (<https://doi.org/10.1101/2023.06.02.543446>) reporting Kv1.2 structures in C-type inactivated state amongst structures in other conditions. This does not affect the novelty of the current findings because it is a different Kv channel and has the focus on the more classical changes in the selectivity filter characteristic for C-type inactivation, but it could be included in the citations for other Kv channels in the introduction or discussion.

Author Rebuttals to Initial Comments:

Referee #1 (Remarks to the Author):

A. Summary of the key results

Fernández-Mariño and colleagues have identified a novel inactivation motif in Kv2.1 channels. This site, comprised of four nonpolar residues coming together at the intracellular end of S6, appears to be distinct from those involved in traditional N-type and C-type inactivation processes. The identification was achieved through the combined use of electrophysiological functional assays and cryo-EM structural methods, applied to both the wild-type and an inactivation mutant embedded in nano-discs.

B. Originality and significance

The methodologies and approaches utilized in this study are well-established and the discovery of the so-called "hydrophobic nexus" is notable. I concur with the authors' assertions regarding the potential ramifications of this study. Analogous mechanisms and motifs may exist in other voltage-gated ion channels, including voltage-gated sodium and calcium channels. The intimate relationship between the hydrophobic component and the S4-S5 linker may also suggest how certain voltage-gated ion channels can recover from inactivation without needing to reopen. If this proves correct, it will represent a significant advancement in our understanding.

While some issues remain unresolved, the core messages intended by this study are quite noteworthy. Ideally, the authors would have elucidated the F412L structure such that the correlation between function and structure would have been firmer. However, even in the absence of the F412L structure or the homomeric L403A structure (bearing in mind the potential uncertainty regarding whether the structure truly depicts a stable inactivated state corresponding to functional measurements), the overall messages appear convincing and interesting enough. I am quite intrigued by the proposal that the IFM segment of voltage-gated Na⁺ channels impairs the nexus interaction and allows the inactivation process to occur.

C. Data & methodology: validity of approach, quality of data, quality of presentation

The approach and the data quality are fine. There may be a degree of inconsistency regarding which experimental conditions are used (data traces with 2 mM K⁺ and other concentrations); the authors could better justify those choices made. Overall, the approach is straightforward and I do not see any concerns.

We have added a section to the methods to give the reader a better sense for how we chose different external K⁺ concentrations for different experiments.

D. Appropriate use of statistics and treatment of uncertainties

I see no issues here. I appreciated that the authors refrained from null hypothesis testing. It would be beneficial if they showed all data points in addition to the mean and SEM values. For example, there are multiple bar graphs where the authors can easily superimpose all individual data points. Also, the authors could probably state what, for example n = 3 means – 3 eggs from one donor/one RNA injection, etc.

For the two bar graphs presented in the original Extended Data Fig 5 L and M (Now Extended Data Fig 3), we have added individual data points and we now define n explicitly in the methods section. We also show individual data points to the new Extended Data Fig. 2d and Fig. 5i. For all experiments, oocytes were obtained from at least two frogs (and up to 10) and values for n represent the number of oocytes studied (provided in the figure legends). We have added this information to the methods.

E. Conclusions: robustness, validity, reliability

The overall messages are strong. However, the terminology the authors coined – “hydrophobic coupling nexus” – might need reconsideration. Please see below for more.

F. Suggested improvements (and questions)

Specific suggestions listed. Some are really minor. Some are subjective personal bias issues.

We really appreciate the supportive and thoughtful comments of the reviewer and we have tried to address all their concerns or suggestions.

- “Hydrophobic coupling nexus”

From the mutagenesis results, it is evident that select nonpolar residues, namely F412, L316, L329, and L403, play crucial roles. Both F and L are typically considered to be nonpolar and perhaps hydrophobic. The term “hydrophobic coupling nexus” implies (and the authors also suggest?) that hydrophobic interactions among them are indeed essential to prevent inactivation. In this context, the more complete inactivation phenotype in the mutant F412L seems puzzling to me if I just consider only hydrophobic interactions. The electrons in the F aromatic side chain are more delocalized than the electrons in the L aliphatic isobutyl side chain; the L side chain should be more hydrophobic? In my review of the structure files provided, the F412 side chain is stabilized by the Y416 side chain in the wild-type channel and also in 3 of the 4 subunits in the L304 channel – do I have this right?. These F412-Y416 interactions are not totally straight-on face-to-face interactions but they are slightly off-centered staggered interactions. My understanding is that such slightly staggered interactions are quite common. In the mutant protomer D, this arrangement is notably impaired. Thus at the end, it may not be the hydrophobic interactions that are important for keeping the channel away from inactivation but it may turn out to be this pi-pi interaction between F412 and Y416. As a result, “hydrophobic coupling nexus” might not be the best term. Something like “internal coupling nexus”? As presented in this manuscript, the exact mechanisms are not that clear.

In addition, the above observations, if correct, suggest that some mutations encompassing F412 and Y416 may be in order – at least, I would like to see them (and I expect that many readers would also).

These are all excellent points and we appreciate the reviewer taking the time to look at the structure carefully. We have chosen to refer to F412 and its interaction with three surrounding Leu residues as the hydrophobic coupling nexus because their interactions would be expected to be dominated by hydrophobic interactions. It is true that the F412L mutation doesn't decrease the hydrophobicity of this nexus, but it does change the size of the F412 side chain and would be expected to weaken the interaction between these residues, as we see in the L403A mutant structure. F412 and the residue the reviewer mentions (Y416) are located at the S6 helix and can be denoted as (i, i+4) in terms of their positions along the helix. Their backbone atoms form a hydrogen bond (~17 kJ/mol) to stabilize the helix structure. However, the formation of the helix restricts the sidechain entropy, particularly when two ring-containing residues are present at positions (i, i+4), indicating that there is an interaction between the sidechains of F412 and Y416. The calculated free energy of the sidechain interaction is approximately 1.34 kJ/mol, which is considerably smaller compared to the free energy associated with Van der Waals interaction (~6 kJ/mol) or hydrogen bonds. As the hydrophobic coupling nexus will form more Van der Waals interactions, it would be expected to play a dominating role on sidechain entropy of F412. Additionally, the closest distance between the two sidechains of F412 and

Y416 is approximately 4.4Å (between C δ 2 of F412 and C β of Y416), while the distance between their centroid of aromatic rings is around 5.9Å, suggesting that the hydrophobic interaction between the two residues is likely to be weak. While the two aromatic side chains display π - π interaction in stagger stacking manner, the energy associated with this interaction is estimated to be around 0.83 – 1.67 kJ/mol, considering that the centroid distance between the aromatic rings is 5.9 Å. In the mutant protomer D, the displacement of F412 from the hydrophobic coupling nexus leads to an elevation in the sidechain entropy, consequently resulting in an increased sidechain entropy for Y416. The mobility disrupts the density of the sidechain, so that we can only accurately determine the backbone conformation as helical based on the unsharpened map (see Extended Fig. 6f). As the interactions between F412 and the three Leu residues are likely dominated by hydrophobic forces, we would like to retain the name 'hydrophobic coupling nexus'.

- “allosteric mechanisms of inactivation”...

The term “allosteric” is so abused and misused. It would be much if the authors could avoid this (unless this term is specifically defined and used in the manuscript).

We completely agree with the reviewer and have removed the term allosteric from the manuscript in the three places where it was used in the original submission.

- Page 4 “The structure of Kv2.1 ... are conserved”. This sentence seems a bit odd to me. The structure reveals that the residues are conserved?

We have rephrased this sentence.

- Fig. 4A. Perhaps, highlight R308 in the figure also?

R308 has been added to the figure.

- Page 4 “... that predominates at 0 mV...most of the charges accessible to the external solution”. Some readers may be confused based on the GV data in Fig. 1A. The relative G value there is only 0.5. Also it may be helpful to remind the readers about the previous estimates of the number of equivalent gating charges involved in Kv2.1.

Good point. We have revised this section as suggested to point out that this comes from the Q-V data where most of the charge has moved at 0 mV, and we have tried to make this section of the manuscript more accessible to the general audience. We are unaware of any good measurements of the equivalent gating charge in Kv2.1, except from limiting slope measurements by Tagliatela and Stefani soon after the channel was cloned, which we now cite as ref. 38.

- Fig. 2B Data sweeps. Some readers may skip reading the legend. It may be worthwhile to point out the data sweeps are recorded with 100 mM K+ outside.

We agree that this would be a good idea. We attempted to do this while abiding by the space constraints for the figures and it makes the figures look quite cluttered. We have added justification for using different external K+ concentrations in the methods and we have tried to make sure this is mentioned prominently in each of the legends.

- Fig. 2D F412L data sweeps. Some may want to see data sweeps with a higher concentration of K⁺ (especially in comparison with C-type inactivation).

We see very similar gating currents in high external K⁺ and we have added that data to Extended Data Fig. 4a,b). Incidentally, this same manipulation does not rescue ion conduction in the W434F mutant of Shaker as the influence of the mutant on C-type inactivation is so strong (see Yang, Yan and Sigworth, 1997).

- Page 6 “Our proposed mechanism of inactivation in Kv2.1 ...” This seems somewhat out of place to me. Exactly what the inactivation mechanism is has not been fully explained prior to this sentence.

We felt that the subsequent mechanistic experiments would be easier to follow if we hypothesized our mechanism early on, which we did in the preceding paragraph. Nevertheless, we have reworded this section to hopefully not be quite so awkward.

- W365F appears to activate more slowly also. Any guess as to why?

This mutant was challenging to study and did not express well, so we haven’t explored it in detail except to show that it doesn’t speed slow inactivation like it does in Shaker. As stated in the discussion, we do believe the filter and internal pore are coupled so mutations in the filter could have interesting effects on both activation and inactivation. We will look into this more in subsequent studies.

- Fig. 4B time scale bar labels. Among the four sweep sets, the Shaker-W434F sweeps (top left) look the fastest. But Fig. 4C says that only the W434F+TEA sweeps look slower. I am confused.

We apologize for the confusion, part of which resulted because we used different time scale bars for Shaker W434F and Kv2.1 F412L and by the fact that the Shaker mutant expresses somewhat better than the Kv2.1 mutant. We have remade this figure to use the same time scale bar and selected a better example for Shaker W434F to illustrate the striking difference between the two. We also analyzed a few additional recordings and replotted the population data to be easier to comprehend.

- The L403A structure. I would like to know (for my own education) why the wild-type structure at the end was refined with C4 symmetry and the L403 structure at the end was made with C1 symmetry. The C1 symmetry constrained worked fine for L403, revealing that one of the subunits had a different conformation. How did the decision to use the C1 symmetry option come about?

We always compare refinement with C4 and C1 symmetry constraints and if the protein is symmetric the refinement with C4 will yield higher resolution. In the case of L403, the conformational change is relatively small compared to the overall structure. When performing 3D classification focused on the whole structure or the transmembrane region using C4 or C1 symmetry in programs like RELION or CryoSparc, these programs may not be able to effectively separate the particles adopting a conformation similar to the conducting conformation seen in WT channels from the closed-pore conformation that we eventually detected in the L403A mutant. Instead, they merge them together into one class similar to the conducting conformation seen in the WT channels.

To address this challenge, we first aligned the particles as accurately as possible by imposing C4 symmetry to ensure that they are properly oriented. Next, the particles are expanded using C4 symmetry, with each

particle generating four rotated copies by 90 degrees along the central axis (see diagram). It is important to note that the expanded particles dataset can only use C1 symmetry for 3D classification without image alignment. To distinguish between the two types of particles (WT and inactivated conformations), a mask containing only the pore region was applied during the 3D classification. This pore region mask helps RELION to better differentiate and classify the particles based on their conformational differences.

By employing these strategies, we aim to gain insights into the distinct conformational states of the L403 mutant.

- Page 30 "... in 150 mL recording chambers...". Is this really correct?

Thank you for catching this important typo. It should read μL .

- Extended data Fig. 3. It may be useful remind what the cylinders and colors are in A and state what the three curves in B (masked, not masked, etc?).

Good points. We have revised this legend as suggested for new Extended Fig. 1 and 6.

- Extended data Fig. 5. Did you see any use dependence / holding voltage dependence for Cu-Phe and Cd? It may be good to say how often pulses were applied.

We didn't look at the use dependence carefully enough to draw conclusion about this interesting issue but we have added the pulse frequency to the legend so the reader can see that the cell membrane was held at -90 mV most of the time and only depolarized once every 5 seconds.

- Extended data Fig. 5 GV curves – smaller and shallower components at depolarized voltages. What are those components? Such components are much less noticeable in Fig. 1.

Kv2.1 does tend to have a shallower component in the G-V relation at depolarized voltage that is most noticeable at elevated external K^+ concentrations where tail currents are inward. Fig. 1 uses 2 mM external K^+ and outward tail currents were measured whereas Extended Fig 5 (now Extended Fig. 3) employs 50 mM external K^+ and tail currents are inward.

- Extended data Fig. 6. The WT data traces in Fig. 1 were recorded with 2 mM K^+ outside. If the mutant

sweeps with 2 mM K⁺ outside are presented, comparison with the WT data would be easier (I am not a big fan of using 2 mM K⁺ outside with oocytes, however).

We have added traces for WT with 100 mM external K⁺ to this figure (now Extended Fig. 2) to make comparisons easier.

- Extended data Fig. 7B. The voltage dependence reminds me of “U-type” inactivation reported for Kv1.5.

U-type inactivation has also been studied in Kv2.1 by the Jones laboratory in papers we cite. The interpretation in those studies was that inactivation of Kv2.1 is favored in closed states near the open state compared to the open state or most resting closed state. We suspect that the mechanism of inactivation we describe in this manuscript is likely present in most if not all Kv channels, in some cases along with C-type and N-type inactivation, and it is possible that the inactivation mechanism we describe for Kv2.1 also plays a more dominant role in Kv1.5. It will be interesting to see if we can look into these possibilities in subsequent studies.

- Extended data Fig. 9. Given what is shown in A, I think it would be good to make all the data traces recorded with 4-AP red (D, G, J).

Good point. We have revised the figure accordingly.

- Extended data Fig. 12B. Please state what the curves are.

Thanks for catching this omission. We have revised this legend as suggested.

G. References: appropriate credit to previous work?

The referencing appears adequate (although the breadth of the field may leave room for differing opinions). There are a few formatting issues that need to be cleaned up at a later stage.

We have carefully reconsidered reference of other work in the field to tried and be fair, but as the reviewer points out this is not an easy task given the vast literature on gating mechanisms in Kv channels. We hope the changes have improved the referencing and we have corrected all the formatting issues.

H. Clarity and context: lucidity of abstract/summary, appropriateness of abstract, introduction and conclusions.

Please see my comments about the terms “hydrophobic coupling nexus” and “allosteric mechanisms”. The final conclusions and implications sections might benefit from slight truncation.

We have removed allosteric and shorted the final section as suggested.

Referee #2 (Remarks to the Author):

The manuscript “Structures of the Kv2.1 channel and mechanism of inactivation through electromechanical coupling” by Fernández-Mariño and colleagues describes the novel structure of the wild-type Kv2.1 voltage-activated potassium channel; emphasizing the results description and discussion on the transmembrane domain, since the cytoplasmic domain was not resolved. By mapping mutations causing epileptic encephalopathies in the new channel structure, the authors spotted a region that couples the voltage sensor with the pore. This mechanism seems to be distinct to what it has been

described for the Shaker Kv channel.

I found their results very interesting, especially because the authors demonstrate that Kv2.1 and Shaker have commonalities and differences. For instance, these channels S4 helices move similar distances, as determined by electrophysiology and disulfide bonds, but the C-type inactivation described for Shaker does not apply to Kv2.1, as determined by electrophysiology. In the spirit of this comment, I would suggest toning down the discussion that compares Kv2.1 with sodium and calcium channels. The reason for this is twofold: 1) Not all the structures that the authors compared with were obtained in lipid nanodiscs. Hence the absence or presence of lipids could favor different helices conformations. 2) There are no experimental results on this manuscript to support that calcium or sodium channels undergo a similar electromechanical coupling.

We appreciate the supportive comments of the reviewer and we have worked to qualify the implications for inactivation in Nav and Cav channels. We agree that much more work needs to be done to understand the relationships between inactivation mechanisms in these channels, but we also excited about the possibility that a mechanism like what we propose for Kv2.1 can explain otherwise enigmatic findings for those channels.

In the introduction section:

- Rewrite clarify this sentence in the introduction: "In response to sustained membrane depolarization, Kv channels inactivate, decreasing the flow of ions and critically influencing their contributions to electrical signaling constituting a form of short-term memory".

We have revised this awkward sentence, thank you.

In the results section:

- What is the size of the FL protein?

853 residues, which we have added.

- Extended Figure 1 should show which regions of the protein were cut for purification

We have added a cartoon to this figure to illustrate the regions removed.

- Define: VAPA/B 38-40 and MSP 1E3D1.

Reference to the VAPAs has been removed and MSP defined in the methods.

- I would suggest quantifying excellent: "and the maps were relatively uniform and high quality throughout, with excellent density"

We have reworded as suggested and refer the reader to new Extended Data Fig. 1 where quantitative information and maps are shown.

- Extended data figure 3E seems to be cropped.

Fixed.

- Try to be more quantitative in your descriptions, for example in the following sentence define what well-resolved means “some of which are particularly well-resolved (Fig. 1C,D)”.

We appreciate this comment and have tried to improve this aspect of the description throughout.

- Make sure that the figures are cited in order.

Fixed.

- Add previously identified in a prokaryotic K channel in this sentence: “There are also strong EM densities at four positions with the filter along the central axis (Fig. 1E), suggesting that the filter is occupied by ions at the four sites previously identified 46”.

Done.

- This sentence needs to be rewritten: “Although each of these epileptic encephalopathy mutations are interesting, illustrating how disease can result from both loss and gain of function in Kv2.1, we were inspired to study the F412L mutant in greater detail because its non-conducting phenotype is reminiscent of a mutation in the Shaker Kv channel 53,54 that enabled the mechanism of slow C-type inactivation to be elucidated 13,27,28.”

We have revised the sentence to be easier to read.

- What does optimally mean in the following sentence? Is this in comparison to other channels? “the hydrophobic interactions within the nexus involving F412 appear to be optimally positioned for movements of...”

We have rephrased to be more precise.

- The cartoons on figure 6 do not help highlighting the difference between the wt and the mutant structure. Figure 5I-J does a great job because the differences are striking. Maybe there is no need to have a cartoon, and rather use the space to move some of the many extended figures to the main manuscript.

We struggled with how best to depict the mechanism conceptually. Given these comments and those of reviewer 3, we have removed the figure from the manuscript

Referee #3 (Remarks to the Author):

The manuscript by Fernandez-Marino et al. describes a comprehensive study of the rat Kv2.1 voltage-gated potassium channel which plays an important role in the central nervous system and mutations of the human Kv2.1 channel have been associated with epileptic encephalopathy. The authors utilize a combination of cryo-EM and electrophysiology to characterize a C-terminal truncated, but fully functional Kv2.1 variant and a range of disease mutations which lead to the identification of interesting phenotypes with accelerated inactivation for several mutants located within a hydrophobic nexus formed between neighboring S6 helices and the S4/S5 linker. This region is known to be important for electromechanical coupling and conserved in most domain-swapped voltage-gated cation channels. Extensive

electrophysiological characterization (including gating current measurements, Cys engineering for disulfide/Cd²⁺ Hg²⁺ trapping to manipulate S4 movements, as well as measurements in the presence of TEA and 4-AP to sense the status of the internal pore-flanking S6 gate) is carried out to characterize F412L and other Kv2.1 disease mutants located in the hydrophobic nexus region. It is particularly striking that the F412L mutant can be rescued from being nonconductive by adding 4-AP –which counteracts inactivation by slowing the closure of the lower S6 gate. Furthermore, it is shown that the W365F mutation of Kv2.1 corresponding to the well-studied W434F shaker mutant remains conducting and inactivates more slowly than wild-type – a phenotype which is dramatically different from the shaker W434F phenotype with accelerated inactivation. Together, these functional results provide strong support for the interpretation that the nexus region plays a key role for the mechanism of inactivation of Kv2.1 channels which is distinct from classical C-type activation of Shaker channels and involves dynamic changes in the electromechanical coupling rather than changes in the selectivity filter. The probably most compelling evidence for this new model stems from a cryo-EM structure of the L403A mutant within the nexus of Kv2.1 which exhibits an interesting asymmetrical channel conformation with two subunits having distinct changes in S6 compared to wild-type. This results in removal of the kink in the PXP region (characteristic for the open conformation) and causes obstruction of the pore in the lower gate region, hence explaining the fast inactivating phenotype of this variant. The structure of the selectivity filter and voltage sensor domain are largely unaltered by the mutation. In the discussion, the authors present useful comparisons to the respective regions in Kv4 channels and to Cav/Nav channels. It is suggested that the N-type inactivation of sodium channels is rather an allosteric effect of the IFM motif which inserts itself into the equivalent region of the hydrophobic coupling nexus in Nav1.4 channels (as opposed to the previously proposed mechanism which assumed that the IFM motif directly blocks the pore) and the potential therapeutic relevance for small molecule drugs modulating inactivation kinetics is highlighted.

All in all, the presented work is expertly done and represents a thorough investigation of a medically highly important ion channel and these novel findings may have further implications for the mechanistic understanding of the broader family of voltage-gated cation channels. Electrophysiological experiments are carried out with sufficient n for meaningful statistical analysis and all data points have error bars representing S.E.M which is appropriate for the studies. Accuracy of the fit for the curves is also indicated for all biophysical parameters determined from the electrophysiological data. The cryo-EM maps and structural models are of sufficient quality to support the interpretation of the structural data. The study is well-written and the conclusions are clear and well supported by the experimental results. Therefore, the manuscript is suitable for publication in Nature and I have only a few minor points that should be addressed prior to acceptance of the manuscript:

We greatly appreciate these supportive comments and the time the reviewer took to digest all the information we tried to convey in this manuscript.

Minor points:

1.) Please add a scale bar in the cryo-EM micrographs shown in Extended Figures 2 and 11

Done.

2.) Page 4, top paragraph: “captured in an active state that predominates at 0 mV” -> looks more like 50/50 between open/closed, since the midpoint potentials shown in Figure 1B are close to 0 mV (0.8 for full length Kv2.1 and 1.3 mV for the truncated EM construct)

We have clarified the rationale here more precisely in the revised manuscript as described in response to reviewer 1.

3.) Page 5 and Fig 1F Ext Figure 4 C "...shows key basic residues that have been found to influence the single channel conductance" -> no individual residues shown, just a surface colored according to electrostatic potential. Adjust the Figure or change "residues" to "region"

Due to space constraints and citation limits, we removed this short section of the results.

4.) Page 7, middle: "In the case of L403, the extent..." -> L403A

Fixed.

5.) Page 7 and Figure 4 D, E, Ext. Fig 8: The analysis of heteromeric channels formed between wt and F412L mutant is interesting. It is however not mentioned whether the disease mutation is heterozygous and exhibits a dominant negative phenotype? This would make sense. Include a reference if this has been published.

The F412L mutation is autosomal dominant but whether it is a dominant negative has not been demonstrated. We now comment on the dominant nature of the mutation in the results section.

6.) Figure 6 legend: "Cartoons representing the open state and a symmetric inactivated state" -> why is a symmetric inactive state shown if the structure of L403A has an asymmetrical arrangement of the four subunits and the authors state that there would be clashes between subunits if a symmetrical channel would be created by superposition of the "inactive" C and D subunits on a and B (Extended Fig 14)? The authors suggest that small rearrangements may reduce these clashes and the channel could relax into a symmetrical state with 4 inactive-conformation protomers without clashes, but without results from a coarse-grained MD simulation supporting this idea or experimental data, a 4-fold symmetrical inactive state is purely hypothetical.

We agree that the four-fold symmetrical inactivated state is completely hypothetical. We had chosen it for simplicity, but since reviewer 1 also commented that Fig 6 was not helpful, we have now removed it from the manuscript.

7.) Method section: please add the concentration of Kv2.1 nanodisc sample used for making cryo-EM grids. It currently states only "concentrated samples of Kv2.1 or the L403A mutant in nanodiscs (3 μ L) were applied to glow-discharged Quantifoil grid"

The concentration of both proteins has now been added.

8.) Extended Figure legends are sometimes a bit short and would benefit from more information, eg. Extended Fig. 3 D and Figure 12 B show FSC curves (from Relion?) but there is no legend for the three traces in red, green and blue. I suppose this is masked (bold black) and green is unmasked and red is phase randomization.

We apologize for these omissions and have systematically gone through the figure legends of these figures to make sure all necessary information is provided.

Red curve = rIn Corrected FSC Phase Randomized Masked Maps.

Green curve = rln FSC Unmasked Maps.
Blue curve = rln FSC Masked Maps.
Black curve = rln FSC Corrected.

9.) Extended Figure 16: marking different channel regions on top of the sequence alignment (eg. S4/S5 linker, VSD, S6 etc) would make this figure easier to understand.

The requested additional labels have been added.

10.) While drafting this report, a preprint from Fred Sigworth and coworkers appeared in bioRxiv (<https://doi.org/10.1101/2023.06.02.543446>) reporting Kv1.2 structures in C-type inactivated state amongst structures in other conditions. This does not affect the novelty of the current findings because it is a different Kv channel and has the focus on the more classical changes in the selectivity filter characteristic for C-type inactivation, but it could be included in the citations for other Kv channels in the introduction or discussion.

The citation has been added.

Reviewer Reports on the First Revision:

Referees' comments:

Referee #1 (Remarks to the Author):

The authors have addressed the suggestions and questions raised during the previous round. I think that everything is just about in order for the next stage.

A. Summary of the key results

See the previous comments.

B. Originality and significance

See the previous comments.

C. Data & methodology: validity of approach, quality of data, quality of presentation

No concerns.

D. Appropriate use of statistics and treatment of uncertainties

The authors have adequately addressed the concerns and suggestions.

E. Conclusions: robustness, validity, reliability

The authors have adequately addressed the concerns and suggestions. Please see below for more.

F. Suggested improvements (and questions)

A couple of suggestions (as before, my subjective suggestions) are listed below.

- “Hydrophobic coupling nexus”

I appreciate the authors' reply. However, I am not sure how the pair-wise interaction energy values that the authors presented in the reply document. Those energy values depend on many factors including the local relative permittivity (thus, ions, water, etc). Also the distance values that the authors show have some “wobble”. A future study may address the relative contribution of the aromatic pi-pi interactions, I expect.

- P5 “... the cooperativity of the final opening transition”. I suggest “the concerted final opening transition”.

- Fig. 1a data traces. The legend says “... from -100 mV to +100 mV (20 mV increments)”. Are all those traces shown? Is it possible that only some of them are shown?

- Extended Data Fig. 5d & f

The time constant values are very ill defined for this data set. This should be indicated clearly either graphically or in the legend.

Author Rebuttals to First Revision:

We appreciate the thoughtful remaining comments of the reviewer and we have made the requested changes where appropriate. Our responses are indicated with blue text.

Referee #1 (Remarks to the Author):

The authors have addressed the suggestions and questions raised during the previous round. I think that everything is just about in order for the next stage.

A. Summary of the key results

See the previous comments.

B. Originality and significance

See the previous comments.

C. Data & methodology: validity of approach, quality of data, quality of presentation

No concerns.

D. Appropriate use of statistics and treatment of uncertainties

The authors have adequately addressed the concerns and suggestions.

E. Conclusions: robustness, validity, reliability

The authors have adequately addressed the concerns and suggestions. Please see below for more.

F. Suggested improvements (and questions)

A couple of suggestions (as before, my subjective suggestions) are listed below.

- “Hydrophobic coupling nexus”

I appreciate the authors’ reply. However, I am not sure how the pair-wise interaction energy values that the authors presented in the reply document. Those energy values depend on many factors including the local relative permittivity (thus, ions, water, etc). Also the distance values that the authors show have some “wobble”. A future study may address the relative contribution of the aromatic pi-pi interactions, I expect.

We completely agree and we plan to study Y416 in a future study, as suggested.

- P5 “... the cooperativity of the final opening transition”. I suggest “the concerted final opening transition”.

We have made the suggested change in wording.

- Fig. 1a data traces. The legend says “... from -100 mV to +100 mV (20 mV increments)”. Are all those traces shown? Is it possible that only some of them are shown?

Yes we only showed sweeps every 40 mV and this is now indicated in the legend.

- Extended Data Fig. 5d & f

The time constant values are very ill defined for this data set. This should be indicated clearly either graphically or in the legend.

The reviewer is correct. In the presence of 4-AP, inactivation of Wt Kv2.1 is extremely slow and not adequately described even with 20 s long depolarizations. In response to this excellent point we have added the following statement to the legend to Extended Data Fig. 5f:

“Inactivation is barely detectable on this timescale in the presence of 4-AP and thus τ values in that condition are poorly defined.”